



# Saccharide composition in atmospheric fine particulate matter at the remote sites of Southwest China and estimates of source contributions

**Zhenzhen Wang[1], Di Wu[1], Zhuoyu Li[1], Xiaona Shang[1], Qing Li[1], Xiang Li[1], Renjie Chen[2], Haidong Kan[2], Huiling Ouyang[3],Xu Tang[3], Jianmin Chen[1,3,*]**

[1] Shanghai Key Laboratory of Atmospheric Particle Pollution and Prevention (LAP3), Department of Environmental Science & Engineering, Fudan Tyndall Centre, Fudan University, Shanghai 200438, China

[2] School of Public Health, Key Lab of Public Health Safety of the Ministry of Education, NHC Key Laboratory of Health Technology Assessment, Fudan University, Shanghai 200032, China

[3] IRDR International Center of Excellence on Risk Interconnectivity and Governance on Weather/Climate Extremes Impact and Public Health, Institute of Atmospheric Sciences, Fudan University, Shanghai 200438, China

*Corresponding author: jmchen (jmchen@fudan.edu.cn)

Phone/fax: (+86) 021-31242298/31242080

Address: Songhu Road 2005, Shanghai 200438, China



**Abstract**. Based on source-specific saccharide tracers, the characteristic of biomass burning (BB) and biogenic emissions to saccharides was investigated in three rural sites at Lincang, where covered with 65% of forest in the southwest border of China. The total saccharides accounted for 8.4±2.7% of OC, and 1.6±0.6% of $PM_{2.5.}$ The measured anhydrosugars accounted for 48.5% of total saccharides, among which levoglucosan was the most dominant species. The high level of levoglucosan was both attributed to the local BB activities and biomass combustion smoke transported from the neighboring regions of Southeast Asia (Myanmar) and the northern Indian Peninsula. The measured mono (di) saccharides and sugar alcohols accounted for 24.9±8.3% and 26.6±9.9% of the total saccharides, respectively, were both proved to be mostly emitted by direct biogenic volatilization from plant materials/surface soils, rather than as byproducts of polysaccharides breakdown during BB processes. Five sources of saccharides were resolved by non-negative matrix factorization (NMF) analysis, including BB, soil microbiota, plant senescence, airborne pollen and plant detritus with the contribution of 34.0%, 16.0%, 21.0%, 23.7% and 5.3%, respectively. The results provide the information on the magnitude of levoglucosan and contributions of BB, as well as the characteristic of biogenic saccharides, at the remote sites of Southwest China, which can be further applied to regional source apportionment models and global climate models.

## 1 Introduction

Biomass burning (BB) and biogenic aerosols are thought to play important roles on air quality, human health and climate through direct or indirect effects (Jacobson et al., 2000; Christner et al., 2008; Pöschl et al., 2010; Després et al., 2012; Chen et al., 2017, Tang et al., 2019). Atmospheric saccharides components have been extensively reported to be originate from biomass burning (natural or anthropogenic), suspended soil or dust and primary biological aerosol particles (PBAPs) (e.g., fungal and fern spores, pollens, algae, fungi, bacteria, and plant debris), as well as biogenic secondary organic aerosol (SOA) (e.g., Rogge et al., 1993; Graham et al., 2003; Jaenicke, 2005; Medeiros et al., 2006; Elbert et al., 2007; Fu et al., 2013). As one of the major classes of water-soluble organic compounds, saccharides in atmospheric aerosols have been detected over urban areas, forests, mountains, and remote marine regions (Pashynska et al., 2002; Yttri et al., 2007; Fu et al., 2009; Burshtein et al., 2011; Jia and Fraser, 2011; Chen et al., 2013; Pietrogrande et al., 2014;Li et al., 2016a, b). It has been



reported that saccharides account for 13-26% of the total organic compound mass
identified in continental aerosols, and as much as 63% in oceanic aerosols (Simoneit
et al., 2004).
Levoglucosan and related anhydrosugar isomers (mannosan and galactosan),
produced from pyrolysis of cellulose and hemicellulose, are considered to be
relatively stable in the atmosphere (Schkolnik et al., 2005; Puxbaum et al., 2007), and
thus have been recognized as specific molecular markers for BB source emissions
(Simoneit et al., 1999, 2000; Fraser and Lakshmanan, 2000; Sullivan et al., 2014; Du
et al., 2015). However, some studies have challenged this knowledge and proved that
levoglucosan alone was not suitable to be a distinct marker for BB in various regions
and periods. Because there were evidences that levoglucosan is also emitted from
non-BB sources (Wu et al., 2020), such as coal burning (Rybicki et al., 2020; Yan et
al., 2018), open waste burning (Kalogridis et al., 2018), incense burning (Tsai et al.,
2010), and food cooking (Reyes-Villegas et al., 2018). It was reported that the
levoglucosan emission contribution of BB sources ranged from 21.3 to 95.9% (Wu et
al., 2021). The current studies in China have reported the value of 2.6-289.1 ng m$^{-3}$
and 11.6-1803.1 ng m$^{-3}$ respectively over Beijing and Wangdu in summer (Yan et al.,
2019), 2.4-1064.1 ng m$^{-3}$ over Shanghai all year round (Xiao et al., 2018), 15.6-472.9
ng m$^{-3}$ over Guangzhou (Zhang et al., 2010), 21.1-91.5 ng m$^{-3}$ over Hongkong (Sang
et al., 2011), 60.2-481.9 ng m$^{-3}$ over Xi 'an (Yang et al., 2012), 36.0-1820.9 ng m$^{-3}$
over Chengdu (Yang et al., 2012) and 10.1-383.4 ng m$^{-3}$ over the Tibetan Plateau (Li
et al., 2019). In north China, high concentrations of levoglucosan is a serious problem
due to drastic enhancement of coal and BB for house heating in winter and autumn
(Zhang et al., 2008; Zhu et al., 2016). The BB pollution might be exacerbated under
unfavorable meteorological conditions, such as in the Chengdu basin (Chen and Xie,
2014). In general, BB is an important source of fine particulate matter and with
notable contribution to OC in China (Zhang et al., 2008; Cheng et al., 2013; Chen et
al., 2017), controls on BB could be an effective method to reduce pollutant emissions.
Recently study reported that BB activities have been reduced in China, because the
total levoglucosan emission of China exhibited a clear decreasing trend from 2014
(145.7 Gg) to 2018 (80.9 Gg) (Wu et al., 2020).
Saccharide compounds including a variety of primary saccharides
(monosaccharides and disaccharides) and sugar alcohols (reduced sugars) have been
measured to estimate the contribution of biogenic aerosols, including fungi, viruses,



bacteria, pollen, and plant as well as animal debris (Simoneit et al., 2004; Jaenicke et
al., 2007). For instance, arabitol and mannitol have been proposed as biomarkers for
airborne fungal spores (Bauer et al., 2008; Zhang et al., 2010; Holden et al., 2011;
Liang et al., 2013a, b), because both of them can function as storage or transport
carbohydrates to regulate intracellular osmotic pressure (Bauer et al., 2008). Glucose
and sucrose are thought to originate from natural biogenic detritus, including
numerous microorganisms, plants, and animals (Simoneit et al., 2004; Tominaga et al.,
2011). As the oxidation products of isoprene, methyltetrols (including
2-methylthreitol and 2-methylerythritol) have been suggested as tracers of
isoprene-derived SOA (Claeys et al., 2004; Kleindienst et al., 2007; Ding et al., 2016).
In the previous study, the contributions of fungal spores to OC were estimated to be
$14.1 \pm 10.5\%$ and $7.3 \pm 3.3\%$ respectively at the rural and urban sites of Beijing
(Liang et al., 2013b). Airborne pollen and fungal spores contributed 12-22% to the
total OC in ambient aerosols collected in Toronto (Womiloju et al., 2003). Jaenicke
(2005) found that PBAPs can comprise from 20 to 30% of the total atmospheric PM
(>0.2 mm) from Lake Baikal (Russia) and Mainz (Germany). However, there is still a
limited number of studies on quantifying the abovementioned biogenic aerosol
contributions to ambient aerosol.
Lincang located in the southwest border of China is a traditional agricultural area of
Yunnan province, where planting a large area of tea, sugar cane, rubber, macadamia
nuts, etc. It is the largest production base of black tea and macadamia nut in China.
The forest coverage rate of Lincang reaches to 65%. It has a wide variety of plant
species, and has 6 nature reserves covering an area of ~222,000 hectares, accounting
for 8.56% of the total area. As a residential area for ethnic minorities, Lincang has
unique culture, humanity and living habits. A high portion of houses with wood
burning used for cooking in villages in proximity and a large area of Southeast Asia
constitute, and forest fires were frequently happened in this area especially in the dry
seasons (March-April). All implies that there are abundant biogenic aerosols in the
area, and BB pollution may be an important potential source of air pollution. However,
little information on the magnitude of biogenic and BB tracers in this area is available.
The contributions of biogenic aerosol and BB, and BB types are poorly understood.
In this study, the sampling were conducted from March 8 to April 8, 2019 at three
mountaintop sites of Lincang, where is an ideal site for investigating the BB emission
characteristics. BB tracers (including anhydrosugars and $K^+$) and biogenic aerosol



tracers (primary saccharides and sugar alcohols) were measured to gain the
information on source and contributions of BB and biogenic emissions in PM$_{2.5}$ over
the rural Lincang. This study would be useful and valuable for provide reliable
information on sources and magnitudes of saccharides involving rural BB and
biological emissions in China.

## 2 Experimental section

### 2.1 Aerosol sampling

The PM$_{2.5}$ samples were simultaneously collected on three mountaintop sites,
respectively of Datian (24.11◦ N, 100.13◦ E, 1960 m asl), Dashu (24.12◦ N, 100.11◦ E,
1840 m asl) and Yakoutian (24.12◦ N, 100.09◦ E, 1220 m asl), in Lincang, Yunnan
Province, China, which are located ~300 km west to Kunming (the capital of Yunnan
province) and ~120 km east from the Burma border. These sites are surrounded by
massive mountains and scattering villages without obvious nearby traffic or major
industry emissions. Each sampling was performed over a 23.5 h period every day, and
was collected on quartz by high-volume air samplers (Thermo) equipped with a size
selective inlet to sample PM$_{2.5}$ at a flow rate of 1.13 m$^3$ min$^{-1}$. Altogether, 91 samples
were collected during 8 March to 9 April in 2019.
Quartz filters (Whatman, 8 × 10 in.) were prebaked at 550 °C for 4 h in a muffle
furnace to remove organic material, and were then stored in pre-baked aluminum foils.
The samples were stored at about −20 °C in a refrigerator until analysis. Field blanks
were collected by mounting filters in the sampler without air flow to replicate the
environmental exposure. The data reported were corrected by the blanks at the
sampling sites.

### 2.2 Measurements

The concentrations of organic carbon (OC) and elemental carbon (EC) were
measured using a Multiwavelength Carbon Analyzer (DRI Model 2015; Atmoslytic
Inc., USA). Typically, a 0.58 cm$^2$ punch of the filter was placed on a boat inside the
thermal desorption chamber of the analyzer, and then stepwise heating was applied.
Carbon fractions were obtained following the Interagency Monitoring of Protected
Visual Environments (IMPROVE-A) thermal/optical reflectance (TOR) protocol
(Chow et al., 2007). Replicate analyses were conducted once every ten samples.
Blank sample was also analyzed and used to correct the sample results.
A punch (4.7 cm$^2$) of each quartz filter was ultrasonically extracted with 10.0 mL of



de-ionized water (resistivity = 18.2 MU) for 40 min. The aqueous extracts were
filtrated through syringe filters (PTFE, 0.22 μm) to remove insoluble materials. Ion
chromatography (Metrohm, Switzerland) coupling with Metrosep C6-150 and A6-150
columns was used to detect water-soluble ions ($Cl^-$, $NO_3^-$, $PO_4^-$, $SO_4^{2-}$, $Na^+$, $NH_4^+$,
$K^+$, $Mg^{2+}$, and $Ca^{2+}$) with a detection limit (DL) range of 0.001-0.002 μg m$^{-3}$.
Five saccharide alcohols (glycerol, erythritol, inositol, arabitol and mannitol) and
five primary saccharides (fructose, glucose, mannose, sucrose and trehalose), together
with three anhydrosugars (levoglucosan, mannosan and galactosan) were quantified
by an improved high performance anion-exchange chromatography system coupled
with a pulsed amperometric detector (HPAEC-PAD) (Engling et al., 2006; Caseiro et
al., 2007; Zhang et al., 2013). This method developed by Engling et al. (2006) was
validated to be a powerful method for the detection of carbohydrates without
derivatization techniques, and has been successfully applied for the atmospheric
tracers (e.g., Zhang et al., 2010; Holden et al., 2011; Liang et al., 2013a, b; Li et al.,
2016a, b; Kalogridis et al., 2018; Yan et al., 2018). The separation of the saccharides
was performed on an ion chromatograph (Metrohm, Switzerland) equipped with a
Metrosep Carb 2-250 analytical column and a guard column. The aqueous eluent of
sodium hydroxide and sodium acetate was pumped by a dual pump module at a flow
rate of 0.4 mL min$^{-1}$. The low concentration of 50 mM sodium hydroxide and 10 mM
sodium acetate (eluent A) was applied to pump 1, while the high concentration of 250
mM sodium hydroxide and 50 mM sodium acetate (eluent B) was applied to pump 2.
The gradient generator was set as: 0-10 min, 100% of eluent A; 10-20 min, 50% of
eluent A and 50% of eluent B; 20-50 min, 100% of eluent B; 50-60 min, 100% of
eluent A for equilibration. The extraction efficiency of this analytical method was
determined to be better than 90% based on analysis of quartz filters spiked with
known amounts of mannitol. The method DL of the referred carbohydrate compounds
were 0.005-0.01 mg L$^{-1}$. All carbohydrate species were below detection limits in the
field blanks.
**2.3 Other data**
The meteorological parameters, including temperature (T), relative humidity (RH),
solar irradiation (W m$^{-2}$), and rainfall (mm) were obtained from the Physical Sciences
Laboratory of NOAA (https://psl.noaa.gov). The temporal changes in meteorological
variables over the observation sites during the sampling periods are shown in Figure



S1.
In order to characterize the origin and transport pathway of the air masses to the
sampling sites, 72 h back-trajectories of the aerosol were calculated using Hybrid
Single-Particle Lagrangian Integrated Trajectory (HYSPLIT) model developed by
NOAA/ARL (Draxler and Hess, 1998) via NOAA ARL READY Website
(http://ready.arl.noaa.gov/HYSPLIT.php) with an endpoint height of 1500 m. To
investigate the influence of BB emissions, fire pixel counts were obtained from
Moderate Resolution Imaging Spectroradiometer (MODIS) observations on NASA
satellites (https://earthdata.nasa.gov/).

### 2.4 Statistical analysis

A Pearson's correlation test was performed using the Statistical Product and Service
Solutions (SPSS) software for the dataset containing ambient concentrations of the
measured saccharides, inorganic ions and solar irradiation. Non-negative matrix
factorization (NMF) analysis was utilized to resolve potential emission source and
estimate their contribution to atmospheric saccharides. NMF introduced by Lee and
Seung (1999) was similar to positive matrix factorization (PMF). Both methods find
two matrices (termed the contribution matrix of W and the source profile matrix of H)
to reproduce the input data matrix (V) using the factorization approach (V = WH) as a
positive constraint (W ≥ 0 and H≥ 0). However, PMF forces the negative factors to be
positive, but NMF method only retains nonnegative factors. NMF minimizes the
conventional least-squares error and the generalized Kullback-Leibler divergence
(Shang et al., 2018). Therefore, the results obtained from NMF are more responsive to
the original characteristics of input data set and less number of factors will be
extracted (Zhang et al., 2019). Half of the DL was used for the value below the
detection limit. In this study, galactosan, mannose and inositol were excluded because
their concentration was mostly below the DL. Concentrations of the other ten
saccharide species for the total 91 samples were subjected for NMF analysis. The
uncertainties in NMF analysis were estimated as 0.3 plus the analytical detection limit
according to the method of Xie et al. (1999). The constant 0.3 corresponding to the
log (Geometric Standard Deviation, GSD) was calculated from the normalized
concentrations for all measured species, and was used to represent the variation of
measurements.

### 3 Results and Discussion



## 3.1 Saccharides concentration and composition


The temporal variations of $PM_{2.5}$ mass, OC, EC, and various saccharides measured
in all samples are shown in Figure 1. A statistical summary of all the data are listed in
Table S1. During the sampling periods, the $PM_{2.5}$ mass concentrations ranged between
13.7-87.8 µg m$^{-3}$ with average values of 41.8 µg m$^{-3}$. The concentrations of OC and
EC varied between 2.5-22.4 and 0.3-4.3 µg m$^{-3}$ with average values of 8.4 and 1.7µg
m$^{-3}$, respectively. OC accounted for 19.9 ±3.7% of total $PM_{2.5}$ mass. The ambient
concentrations of the total saccharides varied between 244.5-1291.6 ng m$^{-3}$ with
average values of 638.4 ng m$^{-3}$. The total saccharides quantified in $PM_{2.5}$ accounted
for 8.4±2.7% (range: 3.8%-20.6%) of the OC, and accounted for 1.6±0.6% (range:
0.6%-3.0%) of the $PM_{2.5}$. Figure 2 presents the mean concentration levels of 12
measured saccharide compounds, categorized into anhydrosugar, sugar and sugar
alcohol, as well as the relative contribution of these saccharides for the all samples.
The values for each site are shown in Figure S2. The absolute concentration and
relative contribution of each saccharide displayed no distinct variation between the
three sites.

## 3.1.1 Anhydrosugars


The mean concentrations of levoglucosan and mannosan were 287.7 and 31.6 ng
m$^{-3}$, respectively with a range of 95.6-714.7 and 0-134.7 ng m$^{-3}$ for all the 91 samples.
Galactosan was only detected in 6 samples, with a range of 2.5-5.5 ng m$^{-3}$. The
anhydrosugars accounted for 48.5% of total measured saccharides. Levoglucosan was
the most dominant specie among all the saccharides. The mean levoglucosan
concentration in this study was comparable to the value at urban Beijing collected in
spring of 2012 (above 200 ng m$^{-3}$) (Liang et al., 2016) and at urban Xi'an collected in
winter of 2015 (268.5 ng m$^{-3}$) (Wang et al., 2018). It was higher than the value at
rural Tengchong mountain site (193.8 ng m$^{-3}$) (Sang et al., 2013), at urban Shanghai
collected in spring of 2012 (66.0 ng m$^{-3}$) (Li et al., 2016) and at urban Hong Kong
collected in spring of 2004 (36.0 ng m$^{-3}$) (Sang et al., 2011), as well as at urban
Beijing collected in summer of 2013 (49.4 ng m$^{-3}$) (Yan et al., 2019), but was lower
than that at a rural site of Xi'an (0.93 mg m$^{-3}$) (Zhu et al., 2017) and at a rural site in
eastern central India (2258 ng m$^{-3}$) (Nirmalkar et al., 2015). During the observation
period, several instances of elevated levoglucosan occurred, peaked on March 8, 16,
23 and April 1. It was thought the ambient levoglucosan were primarily attributed to





domestic biomass fuel burning, the high levoglucosan emission on these peak days
might be from open BB events.
Regression analyses of levoglucosan and the other two anhydrosugars (mannosan,
galactosan) are shown in Figure 3a. Levoglucosan was highly correlated with
mannosan and galactosan, with coefficients of determination (R) of 0.81 (P < 0.01)
and 0.89 (P < 0.01), respectively, indicating similar combustion sources of them. The
ratios of levoglucosan to mannosan (L/M) and mannosan to galactosan (M/G) had
been employed to identify the specific types of BB, although these ratios were quite
variable (Fabbri et al., 2009; Sang et al., 2018). Previous studies suggested that L/M
ratios for burning of softwood were 3-10, hardwood were 15-25, and those from crop
residues were often above 40 (Cheng et al., 2013; Zhu et al., 2015; Kang et al., 2018).
The average L/M and M/G ratios were statistically reported as 32.6 and 1.2 for crop
residues combustion, 4.0 and 3.9 for softwood combustion, 21.5 and 1.5 for hardwood
combustion, respectively (Sang et al., 2013; Shen et al., 2018). In this study, the ratios
of L/M and M/G ranging from 4.7 to 16.1 (mean: 9.7, n=91) and from 3.9 to 6.1
(mean: 4.8, n=6), respectively, crudely indicating major contribution from softwood
burning. The sample collected during 31 March-1 April and during 8-10 March
respectively had considerably lower and higher concentrations of mannosan than
predicted by the levoglucosan-mannosan regression model (Figure 3a). The results
suggested that BB aerosols collected during 31 March-1 April (L/M = 11.52 ± 1.34)
and during 8-10 March (L/M = 6.57 ± 0.53) have originated from different types of
BB as compared with the remaining sampling periods (L/M = 9.34± 1.20). Therefore,
the high levoglucosan emission during 31 March-1 April and during 8-10 March
might be from different open BB events, possibly an open agricultural waste burning
event or a forest fire, whilst the BB of most sampling days originated from biomass
fuel for domestic cooking and heating. It was worth noting that these peak days
neared the Qingming Festival. Another possibility of BB events was that people
burned ghost money to sacrifice ancestor according to Chinese tradition.
Anhydrosugars and water-soluble potassium ($K^+$) have been both widely utilized as
source tracers of BB emissions (e.g., Puxbaum et al., 2007; Wang et al., 2007; Zhang
et al., 2008; Engling et al., 2011). The daily variation on concentrations of
levoglucosan and $K^+$ are shown in Figure S3, the regression analysis of $K^+$ and three
anhydrosugars is shown in Figure 3b. $K^+$ was weakly correlated with levoglucosan,
mannosan, and galactosan, with R values of 0.33, 0.28, and 0.74, respectively. It could



be explained by the additional emissions of $K^+$ from soil and sea water. Since Lincang
is far from the coast, sea salt could be negligible. Because the inhomogeneity of
crustal $K^+$ associated with soil types, it was difficult to fully account for crustal $K^+$
contributions from soil (Harrison et al., 2012; Cheng et al., 2013). The ratio of
levoglucosan to $K^+$ ($L/K^+$) was also used to track possible sources of BB in the
previous studies. The ratios of $L/K^+$ strongly depended on BB processes, namely
smoldering and flaming. Studies suggested that relatively high $L/K^+$ ratios were
obtained from smoldering combustion at low temperatures compared with flaming
combustion (Schkolnik et al., 2005; Lee et al., 2010). Previous results showed the
emissions from the combustion of crop residuals such as rice straw, wheat straw and
corn straw exhibited comparable $L/K^+$ ratios, typically below 1.0. The averages of
$L/K^+$ ratios in this study was $0.48 \pm 0.20$, which was higher than the ratio for corn
straw ($0.21 \pm 0.08$), but was lower than the ratio for Asian rice straw ($0.62 \pm 0.32$)
(Cheng et al., 2013). In this study, higher $L/K^+$ ratios were observed during 8-10
March ($1.20 \pm 0.19$) than those during 31 March-1 April ($0.40 \pm 0.13$), which
suggested that the open fire event during 8-10 March was more possibly due to
smoldering combustion of residues at low temperatures.
Figure 3c and 3d show the scatter plots and regression analyses of $K^+$ versus $PM_{2.5}$,
OC and EC, and levoglucosan versus $PM_{2.5}$, OC and EC, respectively. Linear
regression of $K^+$ on $PM_{2.5}$, OC and EC resulted in R values of 0.64, 0.63 and 0.62,
respectively, which were generally higher than those of levoglucosan on $PM_{2.5}$, OC
and EC, with R values of 0.40, 0.54 and 0.48, respectively. It showed that $K^+$ is more
highly correlated with $PM_{2.5}$, OC and EC. This can be explained by either the
photo-oxidative decay of levoglucosan (Hennigan et al., 2010) and/or different types
of BB processes (Schkolnik et al., 2005; Lee et al., 2010). Even so, the results
supported that the BB posed great impact on fine aerosols. The ratio of levoglucosan
to $PM_{2.5}$ ($L/PM_{2.5}$) is also helpful in distinguishing the contributions of different
levoglucosan sources (Wu et al., 2020). The ratios of $L/PM_{2.5}$ in this study was
0.0041-0.0162 (mean: 0.0072), indicating that levoglucosan emission in the areas
might mainly come from woods (0.01-0.09) and crop straws (0.001-0.008), not
excluding incense burning (0.001-0.007), ritual item burning (0.004-0.086), and meat
cooking (0.005-0.06). However, it is certainly that it was not from corncob burning
(0.0092-0.032), coal burning (0.0001-0.001) and waste incineration (0.0022).
An empirical ratio of levoglucosan to OC (8.2%), calculated from main types of





Chinese cereal straw (rice, wheat and corn) based on combustion chamber
experiments (Zhang et al., 2007), was used to estimate the BB-derived OC. The
average mass concentration of BB-derived OC was 3534.4 ng m$^{-3}$, whilst the
contributions of BB to OC was 41.3%, with a large range of 19.1 to 81.3%. The
contributions are higher than those previous reported, such as 6.5-11% in Hong Kong
(Sang et al., 2011), 18-38% in Beijing (Zhang et al., 2008), 18.9-45.4% over
southeastern Tibetan Plateau (Sang et al., 2013), 26.4-30.2% in Xi'an (Zhang et al.,
2014). The large range of 19.1 to 73.9% revealed that the daily contribution of BB
varied greatly, suggesting open BB event or forest fire happened occasionally.
However, the contribution apportionment of primary BB might be underestimated due
to the degradation of levoglucosan during atmospheric aging of BB influenced air
mass after long-range transport (Hennigan et al., 2010; Mochida et al., 2010; Lai et al.,
2014). Moreover, Wu et al (2020) have reported that the total levoglucosan emission
of China exhibited a clear decreasing trend and biomass burning activities have been
reduced in China. However, it is noteworthy that the mean concentration of
levoglucosan (287.7 ng m$^{-3}$) and the biomass burning contributions to OC (41.3%) at
Lincang mountain site in this study are both higher than the values of 191.8 ng m$^{-3}$
and 28.4% at Tengchong mountain site in 2004 spring (Sang et al., 2013). The result
suggested no significant reduction in BB emissions in Southwest Yunnan Province to
some extent.

### 3.1.2 Mono (di) saccharides

The total concentrations of five mono (di) saccharides, including glucose, fructose,
mannose, sucrose and trehalose, were 25.2-373.7 ng m$^{-3}$ (mean: 158.9 ng m$^{-3}$), which
contributed 24.9±8.3% of the total measured saccharides. The mean values of glucose,
fructose, mannose, sucrose and trehalose were 31.2, 24.6, 2.7, 86.4 and 13.8 ng m$^{-3}$,
respectively. Sucrose was the dominant mono (di) saccharides. The results was
consistent with the previous studies of Yttri et al. (2007), Jia et al. (2010), and Fu et al.
(2012), which had found that sucrose was one of the dominate specie in spring fine
aerosols. The ruptured pollen may be an important source of sucrose in particular of
spring blossom season (Yttri et al., 2007; Fu et al., 2012; Miyazaki et al., 2012). In
spring and early summer, farmland tilling after the wheat harvest causes an enhanced
exposure of soil containing wheat roots to the air, which is beneficial to the release of
sucrose stored in the root (Medeiros et al., 2006), thus resulting in a sharply increased





sucrose.
It was reported that sugars, such as glucose, sucrose and fructose, could be emitted
from developing leaves (Graham et al., 2003). Glucose could be released from both
soils and plant materials (e.g., pollen, fruits and their fragments) (Graham et al., 2003;
Simoneit et al., 2004; Fu et al., 2012). Glucose and sucrose were rich in biologically
active surface soils (Rogge et al., 2007). In this study, the positive correlations were
found between sucrose and glucose (R = 0.52) (Table S2), suggesting a similar origin
of glucose and sucrose in this study. Glucose and fructose have also been identified as
a minor product of cellulose pyrolysis, because they were found to be enrich in BB
emission (Nolte et al., 2001), and correlated well with $K^+$ (Graham et al., 2002) and
levoglucosan (Kang et al., 2018). Herein, no significant correlation were found
between $K^+$, levoglucosan and these mono (di) saccharides. Therefore, the detected
glucose, fructose and sucrose might mostly be emitted by direct volatilization from
plant materials/surface soils, rather than as products of polysaccharides breakdown
during BB processes. The high abundance of sucrose, as well as glucose and fructose
were responsible for biogenic aerosols associated with developing leaves and flowers,
and surface soil suspension.
Trehalose as a stress protectant of various microorganisms and plants (Medeiros et
al., 2006; Jia and Fraser, 2011) was found to be abundant in the fine mode soil, and
has been proposed as a marker compound for fugitive dust from biologically active
surface soils (Simoneit et al., 2004; Medeiros et al., 2006; Rogge et al., 2007; Fu et al.,
2012). Previous study found a positive correlation between trehalose and calcium
(Nishikawa et al., 2000). In this study, there was no significant correlation between
trehalose and calcium. Besides, mannose has been reported to be one of the major
monosaccharide components in phytoplankton, which is originate from marine
biological fragments (Tanoue and Handa, 1987). Mannose was detected in only a few
samples and presented in low concentrations in this study.

### 3.1.3 Sugar alcohols

Five sugar alcohol compounds, including glycerol, threitol, mannitol, arabitol and
inositol were detected in $PM_{2.5}$. These reduced sugars are often reported to be related
with the plant senescence and decay by microorganisms (Simoneit et al., 2004; Tsai et
al., 2013), and are produced by fungi, lichens, soil biota and algae (Elbert et al., 2007;
Bauer et al., 2008). The average concentration of the total sugar alcohols were 159.9





ng m$^{-3}$ with a range of 53.1-254.0 ng m$^{-3}$, which contributed 26.6±9.9% of the total
measured saccharides. Glycerol has been widely found in soil biota (Simoneit et al.,
2004). Herein, glycerol was the second most abundant saccharide with an average
concentration of 123.7 ng m$^{-3}$, which comprised 5.1-44.6% (mean: 22.6%) of the total
measured saccharides.
Mannitol and arabitol have been proposed as tracers for airborne fungal spores
(Elbert et al., 2007; Bauer et al., 2008; Zhang et al., 2010; Burshtein et al., 2011).
Mannitol and arabitol were detected with a concentration range of 0.0- 38.6 ng m$^{-3}$
(14.7 ng m$^{-3}$) and 0.0-21.1 ng m$^{-3}$ (5.8 ng m$^{-3}$), respectively. The mean
concentrations of mannitol and arabitol were comparable to those (mean 11.3 and 9.1
ng m$^{-3}$) reported in the Beijing spring aerosols (Liang et al., 2013b), but were lower
than those (mean 21.9 and 8.43 ng m$^{-3}$) in the Mediterranean summer aerosols
(Burshtein et al., 2011) and (30 and 24 ng m$^{-3}$) at Hyytiälä, Finland in summer (Yttri
et al., 2011). Poor correlations (r = 0.38) were found among mannitol and arabitol in
this study. Nevertheless, a positive correlations was found between trehalose and
mannitol (r = 0.79, P < 0.05) (Table S2).
In the previous studies, the total measured mannitol has been measured and used
for estimating the contribution of fungal spores to organic carbon (Elbert et al., 2007;
Bauer et al., 2008; Zhang et al., 2010). A factor of mannitol per spore (0.49 ± 0.20 pg)
was used to calculate the number concentrations of fungal spores (Liang et al., 2013a),
then the carbon content of fungal spores can be calculated using a conversion factor of
13 pg C per spore obtained earlier as the average carbon content of spores from nine
airborne fungal species, with an uncertainty of 20% (Bauer et al., 2008). The
diagnostic tracer ratio of mannitol to OC was calculated to be 0.0377 according to
these researches (Bauer et al., 2008; Liang et al., 2013a), and then used to estimate the
contribution of fungal spores to the OC. The average mannitol concentrations were
14.7 ± 11.2 ng m$^{-3}$ during the observation period. The average spore-derived OC was
calculated to be 390.3 ng C m$^{-3}$, which contributed of 4.9% of the total OC.
Claeys et al. (2004) firstly identified two diastereoisomeric 2-methyltetrols as
oxidation products of isoprene in the Amazonian rain forest aerosols. Henceforward,
2-methyltetrols has been used as tracers for isoprene-derived SOA (Liang et al., 2012;
Fu et al., 2016; Yan et al., 2019). In the previous studies, erythritol was often
quantified as surrogate of 2-methyltetrols (2-methylthreitol and 2-methylerythritol)
due to lack of standards (Claeys et al., 2004; Ding et al., 2013; Ding et al., 2016). In



this study, concentration ranges of erythritol were 0.4-19.8 ng m$^{-3}$ (mean 11.1 ng m$^{-3}$).
The values of inositol ranged from 0.0 to 22.8 ng m$^{-3}$ with average values of 5.8 ng
m$^{-3}$. Moreover, the sugar alcohols not only originates from biological emissions, but
also derives from BB (Wan and Yu, 2007; Jia et al., 2010). Different levels of glycerol,
arabitol, mannitol, erythritol and inositol in fine particles have been found during
burning of crop residues and fallen leaves as well as indoor biofuel usage for heating
and cooking (Graham et al., 2002; Burshtein et al., 2011; Wang et al., 2011; Yang et
al., 2012; Kang et al., 2018). Herein, only inositol exerted correlation with
levoglucosan (r = 0.42), suggesting inositol may be linked to biomass combustion
sources. Hence, the primarily source of sugar alcohols associated with fine particles
was biogenic aerosols at observation sites.
**3.2 Sources and transport**
Since the distinct concentration of studied compounds was caused by different
emission sources arising from different wind direction, the 72 h backward trajectories
for the samples at Dashu site (24.12∘ N, 100.11∘ E) and the spatial distribution of the
fire spots (March 8-April 8, 2019) were calculated to understand the source of
saccharides in aerosol (Figure 4). The analysis of air mass backward trajectories
suggested that the air mass over Lincang were almost from the westerlies during the
sampling periods, and could be separated into two episodes of remote western source
over 2000 meters and local western source below 2000 meters, as shown in wine red
and green lines. The air masses over 2000 meters were mainly from the regions of
South Asia, such as India and Myanmar. The air masses below 2000 meters were
mainly attributed to the local air flows in the east of Hengduan Mountains.
Mean concentrations of saccharide compounds, as well as the contribution of them,
for the episodes over and below 2000 meters are shown in Figure 5. The mean
concentration of levoglucosan and mannosan for the below 2000 meters samples
(327.4 and 35.6 ng m$^{-3}$) were higher than those for the over 2000 meters samples
(250.3 and 27.3 ng m$^{-3}$). The anhydrosugars accounted for 49.2% and 36.9% of total
saccharides, respectively for the below and above 2000 meters samples. It implied
that the levoglucosan at the observation site was both attributed to the local BB
activities and biomass burning smoke transported from the neighboring regions of
Southeast Asia (Myanmar) and the northern Indian Peninsula. These results were in
agreement with the fact that residents across the Southeast Asia used to utilize woods





as energy source to cook and generate heat.
While for glucose, fructose and sucrose, it was a little higher in the over 2000
meters samples (mean 33.5, 26.4 and 106.2 ng m$^{-3}$) than that in the below 2000
meters samples (mean 29.2, 22.9 and 67.8 ng m$^{-3}$). It implied that biogenic aerosols
(such as ruptured pollen) carrying sugars could pass long distance, which was
supported by previous study, which have observed long-range atmospheric transport
of fine pollen from the Asian continent to the remote island Chichi-Jima under the
influence of westerlies (Rousseau et al., 2008). Although the pollen are usually coarse
with various shapes and hard shells, which results in the relatively short retention time
in the atmosphere. Therefore, it could be concluded that, in addition to the local
pollen, the concentration of sucrose in Lincang was also influenced by the transport of
airborne pollen derived from South Asia areas.

### 3.3 Source apportionment of saccharides

Based on the compositional data of saccharides and key representative markers for
difference sources, five factors associated to the emission sources of saccharides were
finally resolved by NMF. As shown in Figure 6a, factor 1 was characterized by high
level of levoglucosan (71.8%) and mannosan (78.7%), suggesting the source of BB
(Simoneit et al., 1999; Nolte et al., 2001). Factor 2 was characterized by trehalose
(99.9%) and mannitol (100.0%), and was enriched in the other saccharides
components, i.e., arabitol (44.1%), glucose (29.6%), erythritol (18.2%), glycerol
(17.8%), levoglucosan (14.7%), and sucrose (8.6%). These saccharide compounds
had all been detected in the suspended soil particles and associated microbiota (e.g.,
fungi, bacteria and algae) (Simoneit et al., 2004; Rogge et al., 2007). Hence, this
factor was attributed to suspended soil dust and soil microbiota. Factor 3 has high
levels of glycerol (71.4%) and erythritol (58.2%), and showed loadings of glucose
(12.8%) and fructose (11.8%). This factor was thought as the sources from plant
senescence and decay by microorganisms. Factor 4 exhibited a predominance of
sucrose (78.7%), and showed loadings of glucose (17.2%), arabitol (11.8%). This
factor was regarded as the source of airborne pollen, because pollen is the
reproductive unit of plants and contains these saccharides and saccharide alcohols as
nutritional components (Bieleski, 1995; Miguel et al., 2006; Fu et al., 2012). Factor 5
characterized by the dominance of fructose (88.2%) was resolved, and was enriched
in glucose (38.2%) and arabitol (21.2%), thus it could be regarded as the source of



plant detritus.
The pie charts in Figure 6b shows the contribution of each source to total
saccharides. BB of factor 1 (34.0%) was found as the dominant contributor to total
saccharides. Factor 2-5 could all be labeled to a biogenic source accounting for a total
contribution of 66.0%. The sources of soil microbiota (factor 2), plant senescence
(factor 3), airborne pollen (factor 4) and plant detritus (factor 5) respectively
contributed 16.0%, 21.0%, 23.7% and 5.3% to total saccharides. During the sampling
periods, daily variations on proportion of the five factors are shown in Figure S4.
Factor 2 soil microbiota emissions could be associated to soil reclamation and
cultivation of farming periods, and factors 3 plant senescence and factor 5 plant
detritus could be associated to harvesting of vegetation or crop. During the
observation period of a month, along with the weather warming as sunshine enhanced,
human left two obvious traces of cultivated soil during 9-17 March and 27 March-8
April and a trace of vegetation or crop harvest during 17-30 March. The stronger
pollen discharge occurred in March, probably due to the flowering of certain plants.
The BB emissions peaked on 9, 16 March, and 1 April were more prone to be open
burnings.
Since there is still some uncertainty of the factor apportionment, the proportion of
sources are only relative and uncertain. It is still difficult to distinguish the
contributions of microorganisms and plants to biogenic aerosols. Furthermore, all the
above speculations about farming and harvesting periods are based on only one
month's observation, and long-term observations are needed to obtain more accurate
and effective information.

## 4 Conclusion

With the help of the various atmospheric saccharides, this study presents the
characteristic of BB and various biogenic emissions to ambient aerosol in the rural
sites of Southwest China. Levoglucosan was the most dominant specie among all the
saccharides with the concentration of 287.7 µg m$^{-3}$. The ratios of levoglucosan/OC
were 1.9-8.9% (mean: 3.7%). BB contributed to 19.1-73.9% of OC (mean: 41.3%).
The results indicated that domestic biomass fuel burning, open BB events, possibly
open agricultural waste burning, forest fire, or sacrificial activity appeared to be
significant during the spring in this area. The total concentrations of five mono (di)
saccharides and five sugar alcohols respectively contributed 24.9±8.3% and 26.6±9.9%





of the total measured saccharides. Based on the regression analysis, these mono (di)
saccharides and sugar alcohols were mostly emitted by direct biogenic volatilization
from plant materials/surface soils, rather than BB processes. The sampling sites
suffered from both local emissions and BB via long-range transport from Southeast
Asia (Myanmar) and the northern Indian Peninsula. Five sources of saccharides were
resolved by NMF analysis, including BB (34%), soil microbiota (16.0%), plant
senescence (21.0%), airborne pollen (23.7%) and plant detritus (5.3%) at rural
Lincang in spring.
The data herein indicated that biofuel and open BB activities in the rural Southwest
China and neighboring regions could have a significant impact on ambient aerosol
levels. In addition to the residential biofuel usage, field burnings of agricultural
residues and fallen leaves, as well as forest fire, were non-negligible. Some new
technical measures of biomass resource utilization are urgently needed to improve the
open burning emission scenario in rural areas, along with strict prohibition policy of
BB. Meanwhile, the characteristics analysis of saccharides in the region can serve as a
valuable reference for future studies to evaluate temporal variations of biomass
combustion and biogenic emission during modeling predictions and policy making.
**ASSOCIATED CONTENT**
**Supporting Information**
Temporal variations of RH, temperature, solar irradiation and rainfall are shown in
Figure S1. Mean concentrations of saccharide compounds and the contribution of
them for the Datian, Dashu, and Yakoutian samples are shown in Figure S2. Daily
variation on average concentrations of levoglucosan and $K^+$ (a), arabitol and mannitol
(b), $PM_{2.5}$, $Ca^{2+}$ and trehalose (c) at the three sites throughout the sampling period are
shown in Figure S3. Figure S4 showed daily variations on proportion of the five
factors to the total saccharides in $PM_{2.5}$ sampled at three sites during the sampling
periods. Table S1 lists the concentrations of the carbonaceous components and soluble
inorganic ions in $PM_{2.5}$ during the sampling periods of spring 2019. Correlation
matrix for the dataset of the determined saccharides compounds in $PM_{2.5}$ samples is
shown in Table S3.
**Acknowledgments**



This work was supported by the National Natural Science Foundation of China

550     (91843301, 91743202, 91843302).

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



## Captions of Figure and Table

**Figure 1**. Temporal variations of OC, EC, $PM_{2.5}$ and total sugars at the three sites during the sampling periods.

**Figure 2**. The absolute concentration (bar chart) and the relative contribution (pie chart) of various saccharide compounds during the sampling periods.

**Figure 3**. Regression analyses of levoglucosan versus the other two anhydrosugars (a), $K^+$ versus three anhydrosugars (b), levoglucosan versus $PM_{2.5}$, OC and EC (c), and $K^+$ versus $PM_{2.5}$, OC and EC (d).

**Figure 4**. Spatial distribution of the fire spots observed by MODIS, as well as the corresponding 72 h backward air-mass trajectory clusters arriving at 1500 m above ground level during the sampling periods for the collected samples. The backward trajectories were separated into two episodes of remote western source over 2000 meters and local western source below 2000 meters, as shown in wine red and green lines.

**Figure 5**. Mean concentrations and contribution of saccharide compounds for the aerosol samples separated as over and below 2000 meters.

**Figure 6**. Factor profile obtained by NMF analysis (a). Source contribution of the five factors to the total saccharides in $PM_{2.5}$ samples (b).





**Figure 1**

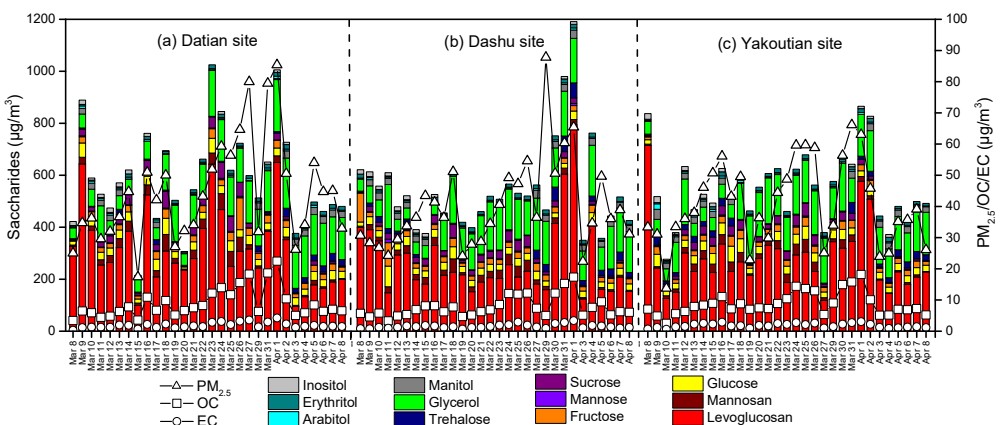

**Figure 2**

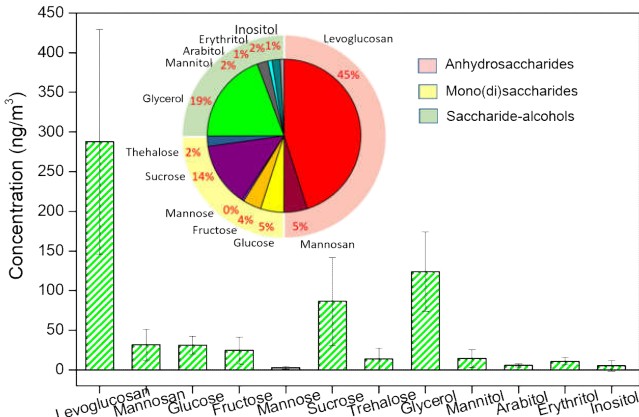





**Figure 3**

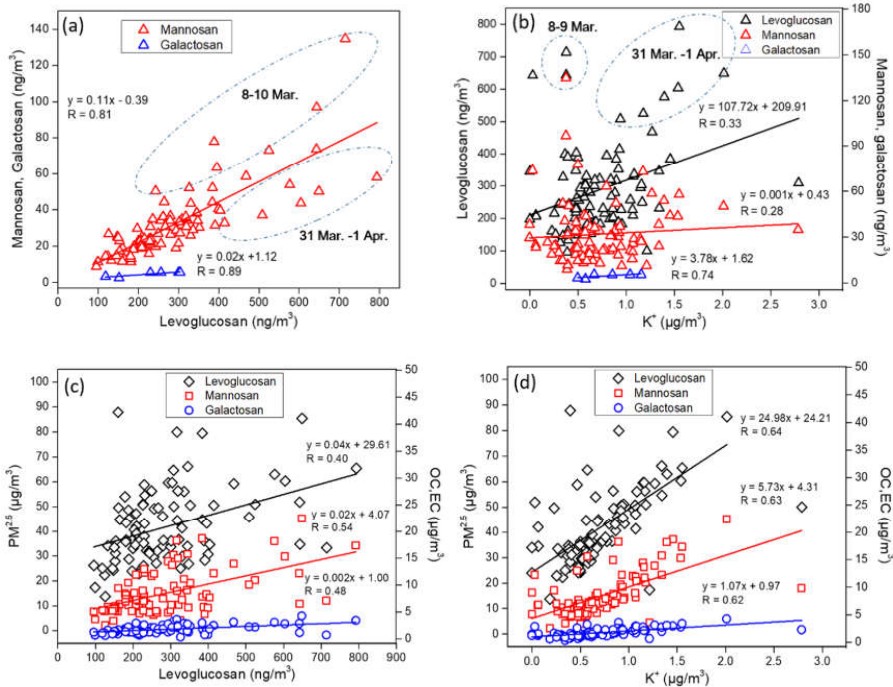



**Figure 4**

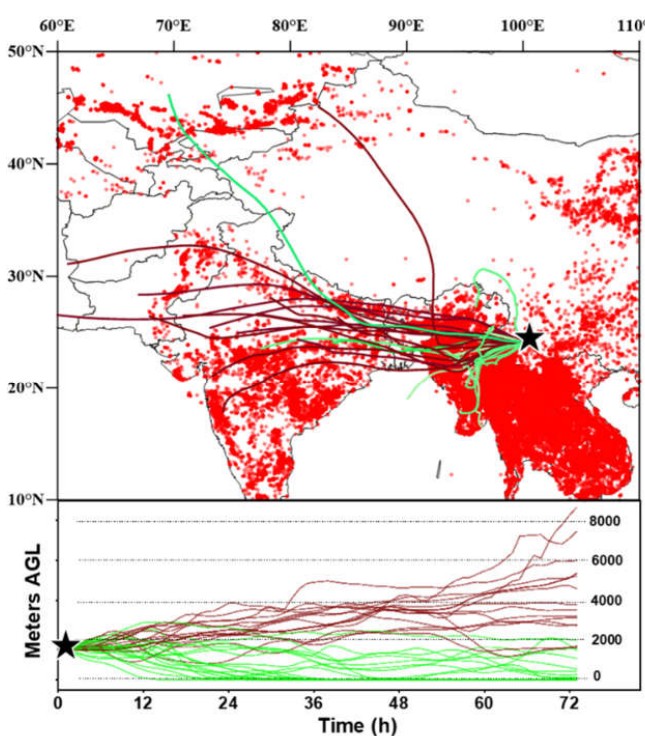



**Figure 5**

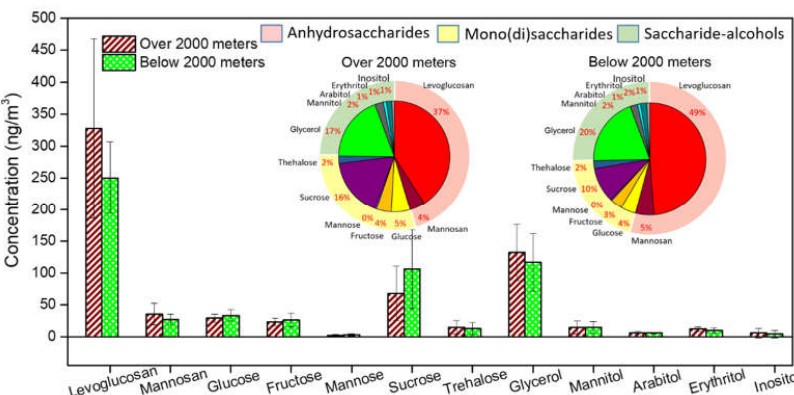



**Figure 6**

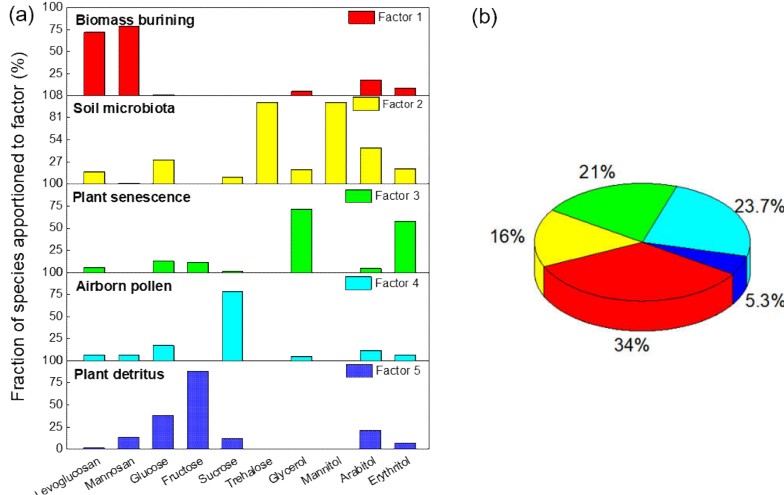