# Peer review of "Saccharide composition in atmospheric fine particulate"

_Atmospheric Chemistry and Physics, 2021_

## Community Comment (CC2)

**General comments:**

In this study the authors reported measurement of $PM_{2.5}$ component over 3 different sites in China during a sampling period of 1 month, during spring 2019. Different saccharides were measured, including biomass burning proxy such as levoglucosan, manossan and galactosan, as well as more uncommon mono(di)saccharide, aiming at tracing the primary biogenic and possibly secondary biogenic sources. After a discussion on the potential link between emissions sources based on correlation and ratio of species, the authors attempt a source-apportionment of the different saccharide using a Non-Negative matrix Factorization (NMF) method and successfully identify 5 different factors of saccharides.

This interesting study reports a comprehensive observational dataset (although not covering the full year) and gives useful insight concerning the sources of organic components thanks to the use of proxy species not-usually used in the literature.

**Reply**:

Dear Prof. Samuel Weber,

We appreciate the positive comments and suggestions about the manuscript. We agree with the reviewer's comments, and have updated the manuscript on the basis of these suggestions.

**Specific comments:**

- Samake et al. (2019) highlight that the different polyols are mostly in the coarse fraction of the PM. Also, it has been hypothesis that the different size distribution of polyols may be a proxy of the different microbiota. Did the authors have also sampled the $PM_{10}$ fraction and could provide the size distribution of the different saccharides?

**Reply:** Thank for the reviewer's suggestion. Indeed, previous results have indicated that polyols (especially mannitol and arabitol) and glucose were prevalent existed in the coarse fraction (Fu et al., 2012; Fuzzi et al., 2007; Pio et al., 2008; Yttri et al., 2007), and were mainly associated with the coarse PM fraction (Samaké et al., 2019). But $PM_{10}$ fraction was not collected due to some practical difficulties, we can't provide the size distribution of the saccharides in this study.

We've cited a reference and rephrased the sentence in line 428-430. **"The contribution of fungal spores might be underestimated because previous results had indicated that mannitol and arabitol were mainly associated with the coarse PM fraction (Samaké et al., 2019)."**

- The source apportionment (SA) is a very interesting part, although it lacks of important information that should be reported: Why didn't you included the whole species available in the SA? It could help identify more robustly BB, but also

saccharides from soil resuspension (with $Ca^{2+}$), and moreover quantify the apportionment of the different factors to the total $PM_{2.5}$ mass.

**Reply:** The source apportionment including the other species could quantify the apportionment of the different factors to the total $PM_{2.5}$ mass. We have tried to include the whole species available in the source apportionment. To make the result be better correlate with the five sources of saccharides, we ran a five-factor NMF. The result is shown as below.

[Figure]

Figure 1. The factor profile obtained by NMF analysis based on the saccharide components (a) and the factor profile based on all the species (b).

In Figrue 1a, the sources of plant detritus (factor 1), plant senescence (factor 2), biomass burning (factor 3), soil microbiota (factor 4) and airborne pollen (factor 5) respectively contributed 5.3%, 21.0%, 34%, 16.0% and 23.7% to the total saccharides. We matched the factors one-to-one in the two figures according to the characteristic saccharide species. The other various species showed decentralized load on these factors. Based on the compositional data of saccharides, five factors associated to the total $PM_{2.5}$ mass were correspond one-to-one to the factors associated to the total saccharides. Factor 1-4 were correspond to the sources of biomass burning, soil microbiota, plant senescence and airborne pollen, respectively. Factor 5 was more appropriate to be thought as a mixed source.

Thus, in Figure 1b, the sources of biomass burning (factor 1), plant senescence (factor 2), soil microbiota (factor 3), airborne pollen (factor 4) and mix sources (factor 5) respectively contributed 16.8%, 28.7%, 13%, 15.8% and 25.7% to the total $PM_{2.5}$ mass. However, we think the naming of these factors associated to the total $PM_{2.5}$ mass are not accurate and comprehensive. In order to get more clear information about the sources and their contribution to the total saccharides, we decided to only report the source apportionment of saccharides.

- It is stated that the SA is still uncertain, but no estimation of the uncertainties is given. It would be of great interest to report the species uncertainties, for instance with bootstraping your input data.

  **Reply:** We only have 91 samples in total, so we cannot carry out resampled runs for many times. The analytical uncertainty was high in present study due to the limited sample number by using the currently used formula in PMF model. We used 0.3 plus the analytical detection limit for estimating uncertainty according to the method of Xie et al. (1999). The constant 0.3 corresponding to the log(Geometric Standard Deviation, GSD) was calculated from the normalized concentrations for all measured species, and was used to represent the variation of measurements. The use of GSD was suitable for our measurement set in a small sample size.

- The timeserie contribution would also be of great interest. Even if the authors did not include a total variable (namely, $PM_{2.5}$), the timeserie of the total saccharide for the 5 factors would be informative.

  **Reply:** We agree with the reviewer's view of the importance on the timeserie contribution. The timeserie of the total saccharide for the 5 factors are shown in Figure S5. We've rewritten the relevant content from Line 525. "During the sampling periods, daily variations on proportion of the five factors are shown in Figure S5. Factor 2 soil microbiota emissions could be associated to soil reclamation and cultivation of farming periods, and factors 3 plant senescence and factor 5 plant detritus could be associated to harvesting of vegetation or crop. During the observation period of a month, along with the weather warming as sunshine enhanced, human left two obvious traces of cultivated soil during 9-17 March and 27 March-8 April and a trace of vegetation or crop harvest during 17-30 March. The stronger pollen discharge occurred in March, probably due to the flowering of certain plants. The BB emissions peaked on 9, 16 March, and 1 April were more prone to be open burnings."

- The "Soil microbiota" factor, identified mainly by the presence of Trehalose and Mannitol (and Arabitol) denotes with the finding of Samake et al. (2020) that found that Arabitol and Mannitol are associated with fungi and bacteria from the leaves and not with the soil (even if some mixing are probable). I would suggest naming it "Soil and leave microbiota".

  **Reply:** We agree with the reviewer's suggestion, "Soil and leaves microbiota" is more specific. We've named it "Soil and leave microbiota" and gave an explanation in line 502-507. "These saccharide compounds had all been detected in the suspended soil particles and associated microbiota (e.g., fungi, bacteria and algae) (Simoneit et al., 2004; Rogge et al., 2007). A recent study found that leaves were a major source

of saccharides-associated microbial taxa in a rural area of France (Samaké et al., 2020). Hence, this factor was attributed to soil and leaves microbiota."

- Overall, the naming of the different factors identified is too rapidly explained, and more detailed could be written to ease the interpretation of the different factors.
  **Reply:** Since each type of sugar has been described in the text, the factors were resolved in a little brief way. In the new version, the naming of the different factors have been more detailed explained from Line 497.

  "As shown in Figure 6a, factor 1 was characterized by high level of levoglucosan (71.8%) and mannosan (78.7%), suggesting the source of BB (Simoneit et al., 1999; Nolte et al., 2001). Factor 2 was characterized by trehalose (99.9%) and mannitol (100.0%), and was enriched in the other saccharides components, i.e., arabitol (44.1%), glucose (29.6%), erythritol (18.2%), glycerol (17.8%), levoglucosan (14.7%), and sucrose (8.6%). These saccharide compounds had all been detected in the suspended soil particles and associated microbiota (e.g., fungi, bacteria and algae) (Simoneit et al., 2004; Rogge et al., 2007). A recent study found that leaves were a major source of saccharides-associated microbial taxa in a rural area of France (Samaké et al., 2020). Hence, this factor was attributed to soil and leaves microbiota. Factor 3 has high levels of glycerol (71.4%) and erythritol (58.2%), and showed loadings of glucose (12.8%) and fructose (11.8%). Kang et al. (2018) reported that glycerol and erythritol presented larger amounts in winter and autumn, when the vegetation decomposed. This factor was thought as the sources from plant senescence and decay by microorganisms. Factor 4 exhibited a predominance of sucrose (78.7%), and showed loadings of glucose (17.2%), arabitol (11.8%). This factor was regarded as the source of airborne pollen, because pollen is the reproductive unit of plants and contains these saccharides and saccharide alcohols as nutritional components (Bieleski, 1995; Miguel et al., 2006; Fu et al., 2012). Factor 5 characterized by the dominance of fructose (88.2%) was resolved, and was enriched in glucose (38.2%) and arabitol (21.2%), thus it could be regarded as the source of plant detritus."

  **Minor comment:**
- Please provide the pie chart of Figure 6b in a non-3D way, as the relative proportion is much harder to see in 3D compare to regular 2D graph.
  **Reply:** We agree with the reviewer's comment. We've provided the pie chart of Figure 6b in a 2D way in the new version of manuscript.

[Figure]

**Figure 6**. Factor profile obtained by NMF analysis (a). Source contribution of the five factors to the total saccharides in PM$_{2.5}$ samples (b).

**References:**

Fuzzi, S., Decesari, S., Facchini, M. C., Cavalli, F., Emblico, L., Mircea, M., Andreae, M. O., Trebs, I., Hoffer, A., Guyon, P., Artaxo, P., Rizzo, L. V., Lara, L. L., Pauliquevis, T., Maenhaut, W., Raes, N., Chi, X., Mayol-Bracero, O. L., Soto-García, L. L., Claeys, M., Kourtchev, I., Rissler, J., Swietlicki, E., Tagliavini, E., Schkolnik, G., Falkovich, A. H., Rudich, Y., Fisch, G., and Gatti, L. V.: Overview of the inorganic and organic composition of size-segregated aerosol in Rondônia, Brazil, from the biomassburning period to the onset of the wet season, J. Geophys. Res., 112, D01201, https://doi.org/10.1029/2005JD006741, 2007.

Pio, C. A., Legrand, M., Alves, C. A., Oliveira, T., Afonso, J., Caseiro, A., Puxbaum, H., Sanchez-Ochoa, A., and Gelencsér, A.: Chemical composition of atmospheric aerosols during the 2003 summer intense forest fire period, Atmos. Environ., 42, 7530–7543, https://doi.org/10.1016/j.atmosenv.2008.05.032, 2008.

Samaké, A., Jaffrezo, J.-L., Favez, O., Weber, S., Jacob, V., Albinet, A., Riffault, V., Perdrix, E., Waked, A., Golly, B., Salameh, D., Chevrier, F., Oliveira, D. M., Bonnaire, N., Besombes, J.-L., Martins, J. M. F., Conil, S., Guillaud, G., Mesbah, B., Rocq, B., Robic, P.-Y., Hulin, A., Meur, S. L., Descheemaecker, M., Chretien, E., Marchand, N., and Uzu, G.: Polyols and glucose particulate species as tracers of primary biogenic organic aerosols at 28 French sites, 19, 3357–3374, https://doi.org/10.5194/acp-19-3357-2019, 2019.

Samaké, A., Bonin, A., Jaffrezo, J.-L., Taberlet, P., Weber, S., Uzu, G., Jacob, V., Conil, S., and Martins, J. M. F.: High levels of primary biogenic organic aerosols are driven by only a few plant-associated microbial taxa, 20, 5609–5628, https://doi.org/10.5194/acp-20-5609-2020, 2020.

---

## Community Comment (CC3)

**General comments:**

The paper entitled "Saccharide composition in atmospheric fine particulate matter at the remote sites of Southwest China and estimates of source contributions" by Zhenzhen Wang and colleagues provide the characteristic of saccharides during spring 2019 at Lincang, a rural site in Southwest China. The authors reported molecule tracers including anhydrosugars, mono (di) saccharides and sugar alcohols, combined with statistical analysis and HYSPLIT model, they concluded that biofuel and open biomass burning (BB) activities could have a significant impact on ambient aerosol levels at Lincang. Overall, this paper is logically organized, and knowledge of this work is needed and helpful for better understanding air conditions in Southwest China. The topic of this paper is within the scope of the journal Atmospheric Physics and Chemistry. I would like to recommend this paper published after the following of my concerns be resolved.

**Reply**: We appreciate the positive comments and suggestions about the manuscript. We agree with the reviewer's comments, and have updated the manuscript on the basis of these suggestions.

**Major comments:**

1.  The surrounding environmental condition is crucial for understanding the results, I strongly suggest the authors added a figure to show the sampling sites as Figure 1. This figure should include some necessary information about the topography, vegetation, residential area nearby Lincang, and photos of three sampling sites are also crucially needed.

    **Reply**: We've added Figure S1 for the location of the sampling sites in the Supporting Information. The number of all the Figures referring to the Supporting Information has been changed.

[Figure]

**Figure S1.** Map of sampling sites. The location of the sampling sites was marked with five-pointed star.

2. The source appointment is mainly based on the 72h backward trajectories of HYSPLIT model. However, high uncertainty existent for the application of HYSPLIT model at high elevation site because topographic relief. The frequencies of HYSPILT or meteorological analysis should provide more creditable results.

    **Reply**: Thank for the reviewer's suggestion. More detailed analyses on topography and meteorology, as well as the frequencies of HYSPILT backward trajectories are stated in the section 3.2 Sources and transport.

    Herein, this sentence has been rewritten. "46.7% of air mass backward trajectories were generally over 2000 meters, while 53.3% of them were below 2000 meters."

    "The southwest wind from the Indian Ocean prevailed at Lincang all the year round. In spring, the southwest wind was often affected by the low temperature downhill wind blowing from the snow-covered Hengduan Mountains. The weather alternated between hot and cold frequently, with unstable air pressure and strong wind. Therefore, the lower air could be diluted by the relatively clean cold air over the plateau. The upper air mainly came from the westerlies."

**Minor comments:**

1. The samples of this work are mainly in spring, the title should be changed to "Saccharide composition in atmospheric fine particulate matter during spring at the remote sites of Southwest China and estimates of source contributions".

    **Reply**: Thank for the reviewer's suggestion. The title have been changed to "Saccharide composition in atmospheric fine particulate matter during spring at the remote sites of Southwest China and estimates of source contributions".

2. Line 62, Wu et al., 2020 is not cited in references.

    **Reply**: Wu et al., 2020 have been cited in Line 62 in the revised manuscript.

3. Line 71-72, "10.1-383.4 ng m$^{-3}$ over the Tibetan Plateau (Li et al., 2019)", the reference Li et al., 2019, EP is glacier cryoconites not aerosol samples.

    **Reply**: "10.1-383.4 ng m$^{-3}$ over the Tibetan Plateau (Li et al., 2019)" have been changed to "10.1-383.4 ng g$^{-1}$ dry weight in cryoconites over the Tibetan Plateau (Li et al., 2019)".

4. Line 75, Sichuan Basin, not "Chengdu Basin".

    **Reply**: "Chengdu basin" have been changed to "Sichuan Basin".

5. Line 79-81, Levoglucosan emission of China is estimated by BB activities by Wu et al., 2021, this sentence is not rigorous.

    **Reply**: This sentence have been rewritten. "Recently study reported that total levoglucosan emission of China exhibited a clear decreasing trend from 2014 (145.7 Gg) to 2018 (80.9 Gg) (Wu et al., 2021), suggesting BB activities might reduce in China.

6. Line 109-112, you should better add some references.

    **Reply**: "Referring to the official website of Lincang Municipal People's Government, the forest coverage rate of Lincang reaches to 65%."

7. Line 116, do you have samples over other period?

    **Reply**: We only sampled at the Lincang sites for a period of about a month.

8. Line 126-130, please add a figure for sample sites.

    **Reply**: We've added Figure S1 for the location of the sampling sites in the Supporting Information.

9. Line 183, why do not use meteorological data at Lincang?

    **Reply**: The satellite data and Lincang meteorological website data were not exactly the same, but were overall similar. In order to obtain more complete data of all indicators, satellite data were used uniformly.

10. Line 231-233, "no distinct variation", has statistical significance?

    **Reply**: Thank for the reviewer's correction. This sentence is not completely accurate. In the revised manuscript, this sentence was deleted.

11. Line 239-248, samples in those references are not collected at the same period.

    **Reply**: Indeed, the samples in these studies were collected at different times. So we presented the specific sampling time of each research. Even if not all

samples were taken in the spring, it would be of great interest to report these information.

12. Line 276-277, how about the L/M for burned ghost money?

**Reply**: "It was worth noting that the peak days during 31 March-1 April (L/M = 11.52 ± 1.34) neared the Qingming Festival. Another possibility of BB events was that people burned ghost money to sacrifice ancestor according to Chinese tradition."

13. Line 290-291, references for L/K$^+$?

**Reply**: We've added the references "(Schkolnik et al., 2005; Lee et al., 2010)".

14. Line 431-441, Figure 4, only one air mass from Hengduan Mountain region. Maybe frequency is better for understanding air sources.

**Reply**: Thank for the reviewer's suggestion. Herein, this sentence has been rewritten. "46.7% of air mass backward trajectories were generally over 2000 meters, while 53.3% of them were below 2000 meters."

15. Line 450-452, how about the atmospheric dynamics for aerosol transport from Southeast Asia to Lincang, especially for residential cooking and heating.

**Reply**: Some sentences were added. "The southwest wind from the Indian Ocean prevailed at Lincang all the year round. In spring, the southwest wind was often affected by the low temperature downhill wind blowing from the snow-covered Hengduan Mountains. The weather alternated between hot and cold frequently, with unstable air pressure and strong wind. Therefore, the lower air could be diluted by the relatively clean cold air over the plateau. The upper air mainly came from the westerlies."

16. Line 512, ng m$^{-3}$?

**Reply**: "μg m$^{-3}$" has been replaced by "ng m$^{-3}$".

17. Line 521, only Myanmar.

**Reply**: "The sampling sites suffered from both local emissions and BB via long-range transport from Southeast Asia (Myanmar, Bangladesh) and the northern Indian Peninsula."

---

## Author Response (AR2)

**Shanghai Key Laboratory of Atmospheric Particle Pollution and Prevention (*LAP³*)**
**Department of Environmental Science & Engineering**
**Fudan University**

Jun. 10, 2021

Dear Prof. Ivan Kourtchev,

Thanks for your kind handling our manuscript!

Here we uploaded our revised manuscript (ACP-2021-83) for consideration to be published on ACP:

**Title**: Saccharide composition in atmospheric fine particulate matter during spring at the remote sites of Southwest China and estimates of source contributions

**Authors**: Zhenzhen Wang, Di Wu, Zhuoyu Li, Xiaona Shang, Qing Li, Xiang Li, Renjie Chen, Haidong Kan, Jianmin Chen

**Special Issue**: The role of fire in the Earth system: understanding interactions with the land, atmosphere, and society (ESD/ACP/BG/GMD/NHESS inter-journal SI)

**Corresponding author:** Jianmin Chen; Address: Department of Environmental Science & Engineering, Fudan University, Shanghai 200433, China; Email: jmchen@fudan.edu.cn.

Thank you for your kind reminder. To better answer the questions from the reviewer #3, we replied to all questions, and also added some sentences reflecting the comments (including questions 1 and 2) to the revised paper.

We appreciate the positive comments and suggestions about the manuscript. We are willing to categorize our manuscript into "Measurement Reports" if necessary.

We acknowledge the comments of three reviewers. The suggestions of the Reviewers gave us great help to improve our manuscript. We have updated the manuscript on the basis of the Reviewers' comments. Below is our response to the comments resulting in a number of clarifications. A marked file in the PDF format was also uploaded so that the reviewers could easily check our update. We expect this manuscript to be published on Atmospheric Chemistry and Physics.

Sincerely yours,

Jianmin Chen

**Comment 1#**

**General comments:**

In this study the authors reported measurement of PM$_{2.5}$ component over 3 different sites in China during a sampling period of 1 month, during spring 2019. Different saccharides were measured, including biomass burning proxy such as levoglucosan, manossan and galactosan, as well as more uncommon mono(di)saccharide, aiming at tracing the primary biogenic and possibly secondary biogenic sources. After a discussion on the potential link between emissions sources based on correlation and ratio of species, the authors attempt a source-apportionment of
the different saccharide using a Non-Negative matrix Factorization (NMF) method and
successfully identify 5 different factors of saccharides.
This interesting study reports a comprehensive observational dataset (although not covering
the full year) and gives useful insight concerning the sources of organic components thanks to the
use of proxy species not-usually used in the literature.
**Reply**:
Dear Prof. Samuel Weber,
We appreciate the positive comments and suggestions about the manuscript. We agree with the
reviewer's comments, and have updated the manuscript on the basis of these suggestions.
**Specific comments:**
1   Samake et al. (2019) highlight that the different polyols are mostly in the coarse fraction of
the PM. Also, it has been hypothesis that the different size distribution of polyols may be a
proxy of the different microbiota. Did the authors have also sampled the $PM_{10}$ fraction and
could provide the size distribution of the different saccharides?
**Reply:** Thank for the reviewer's suggestion. Indeed, previous results have indicated that
polyols (especially mannitol and arabitol) and glucose were prevalent existed in the coarse
fraction (Fu et al., 2012; Fuzzi et al., 2007; Pio et al., 2008; Yttri et al., 2007), and were
mainly associated with the coarse PM fraction (Samaké et al., 2019). But $PM_{10}$ fraction was
not collected due to some practical difficulties, we can't provide the size distribution of the
saccharides in this study.
We've cited a reference and rephrased the sentence in line 440-442. **"The contribution of
fungal spores might be underestimated because previous results had indicated that mannitol
and arabitol were mainly associated with the coarse PM fraction (Samaké et al., 2019)."**
2   The source apportionment (SA) is a very interesting part, although it lacks of important
information that should be reported: Why didn't you included the whole species available in
the SA? It could help identify more robustly BB, but also saccharides from soil resuspension
(with $Ca^{2+}$), and moreover quantify the apportionment of the different factors to the total
$PM_{2.5}$ mass.
**Reply:** The source apportionment including the other species could quantify the
apportionment of the different factors to the total $PM_{2.5}$ mass. We have tried to include the
whole species available in the source apportionment. To make the result be better correlate
with the five sources of saccharides, we ran a five-factor NMF. The result is shown as below.

[Figure]

Figure 1. The factor profile obtained by NMF analysis based on the saccharide components (a) and the factor profile based on all the species (b).

In Figrue 1a, the sources of plant detritus (factor 1), plant senescence (factor 2), biomass burning (factor 3), soil microbiota (factor 4) and airborne pollen (factor 5) respectively contributed 5.3%, 21.0%, 34%, 16.0% and 23.7% to the total saccharides. We matched the factors one-to-one in the two figures according to the characteristic saccharide species. The other various species showed decentralized load on these factors. Based on the compositional data of saccharides, five factors associated to the total $PM_{2.5}$ mass were correspond one-to-one to the factors associated to the total saccharides. Factor 1-4 were correspond to the sources of biomass burning, soil microbiota, plant senescence and airborne pollen, respectively. Factor 5 was more appropriate to be thought as a mixed source.

Thus, in Figure 1b, the sources of biomass burning (factor 1), plant senescence (factor 2), soil microbiota (factor 3), airborne pollen (factor 4) and mix sources (factor 5) respectively contributed 16.8%, 28.7%, 13%, 15.8% and 25.7% to the total $PM_{2.5}$ mass. However, we think the naming of these factors associated to the total $PM_{2.5}$ mass are not accurate and comprehensive. In order to get more clear information about the sources and their contribution to the total saccharides, we decided to only report the source apportionment of saccharides.

It is stated that the SA is still uncertain, but no estimation of the uncertainties is given. It would be of great interest to report the species uncertainties, for instance with bootstraping your input data.

**Reply:** We only have 91 samples in total, so we cannot carry out resampled runs for many times. The analytical uncertainty was high in present study due to the limited sample number by using the currently used formula in PMF model. We used 0.3 plus the analytical detection limit for estimating uncertainty according to the method of Xie et al. (1999). The constant 0.3 corresponding to the log(Geometric Standard Deviation, GSD) was calculated from the normalized concentrations for all measured species, and was used to represent the variation of
measurements. The use of GSD was suitable for our measurement set in a small sample size.
4  The timeserie contribution would also be of great interest. Even if the authors did not include
a total variable (namely, $PM_{2.5}$), the timeserie of the total saccharide for the 5 factors would
be informative.
**Reply:** We agree with the reviewer's view of the importance on the timeserie contribution.
The timeserie of the total saccharide for the 5 factors are shown in Figure S5. We've rewritten
the relevant content from Line 536. "During the sampling periods, daily variations on
proportion of the five factors are shown in Figure S5. Factor 2 soil microbiota emissions
could be associated to soil reclamation and cultivation of farming periods. Factors 3 plant
senescence and factor 5 plant detritus could be associated to harvesting of vegetation or crop.
During the observation period of a month, along with the weather warming as sunshine
enhanced, human left two obvious traces of cultivated soil during 9-17 March and 27 March-
8 April and a trace of vegetation or crop harvest during 17-30 March. The stronger pollen
discharge occurred in March, probably due to the flowering of certain plants. The BB
emissions peaked on 9, 16 March, and 31 March-1 April were more prone to be open
burnings. Therein, the BB during 31 March-1 April was probably from the burning of ghost
money during the Qingming Festival."
5  The "Soil microbiota" factor, identified mainly by the presence of Trehalose and Mannitol
(and Arabitol) denotes with the finding of Samake et al. (2020) that found that Arabitol and
Mannitol are associated with fungi and bacteria from the leaves and not with the soil (even if
some mixing are probable). I would suggest naming it "Soil and leave microbiota".
**Reply:** We agree with the reviewer's suggestion, "Soil and leaves microbiota" is more specific.
We've named it "Soil and leave microbiota" and gave an explanation in line 514-522. "These
saccharide compounds had all been detected in the suspended soil particles and associated
microbiota (e.g., fungi, bacteria and algae) (Simoneit et al., 2004; Rogge et al., 2007). A recent
study found that leaves were a major source of saccharides-associated microbial taxa in a rural
area of France (Samaké et al., 2020). Hence, this factor was attributed to soil and leaves
microbiota."
6  Overall, the naming of the different factors identified is too rapidly explained, and more
detailed could be written to ease the interpretation of the different factors.
**Reply:** Since each type of sugar has been described in the text, the factors were resolved in a
little brief way. In the new version, the naming of the different factors have been more detailed
explained from Line 509.

"As shown in Figure 6a, factor 1 was characterized by high level of levoglucosan (71.8%)
and mannosan (78.7%), suggesting the source of BB (Simoneit et al., 1999; Nolte et al., 2001).
Factor 2 was characterized by trehalose (99.9%) and mannitol (100.0%), and was enriched in
the other saccharides components, i.e., arabitol (44.1%), glucose (29.6%), erythritol (18.2%),
glycerol (17.8%), levoglucosan (14.7%), and sucrose (8.6%). These saccharide compounds had
all been detected in the suspended soil particles and associated microbiota (e.g., fungi, bacteria
and algae) (Simoneit et al., 2004; Rogge et al., 2007). A recent study found that leaves were a
major source of saccharides-associated microbial taxa in a rural area of France (Samaké et al.,
2020). Hence, this factor was attributed to soil and leaves microbiota. Factor 3 has high levels
of glycerol (71.4%) and erythritol (58.2%), and showed loadings of glucose (12.8%) and
fructose (11.8%). Kang et al. (2018) reported that glycerol and erythritol presented larger
amounts in winter and autumn, when the vegetation decomposed. This factor was thought as
the sources from plant senescence and decay by microorganisms. Factor 4 exhibited a
predominance of sucrose (78.7%), and showed loadings of glucose (17.2%), arabitol (11.8%).
This factor was regarded as the source of airborne pollen, because pollen is the reproductive
unit of plants and contains these saccharides and saccharide alcohols as nutritional components
(Bieleski, 1995; Miguel et al., 2006; Fu et al., 2012). Factor 5 characterized by the dominance
of fructose (88.2%) was resolved, and was enriched in glucose (38.2%) and arabitol (21.2%),
thus it could be regarded as the source of plant detritus."

**Minor comment:**
1   Please provide the pie chart of Figure 6b in a non-3D way, as the relative proportion is much
harder to see in 3D compare to regular 2D graph.
**Reply:** We agree with the reviewer's comment. We've provided the pie chart of Figure 6b in
a 2D way in the new version of manuscript.

[Figure]

**Figure 6**. Factor profile obtained by NMF analysis (a). Source contribution of the five factors to the total saccharides in PM$_{2.5}$ samples (b).

**References:**

Fuzzi, S., Decesari, S., Facchini, M. C., Cavalli, F., Emblico, L., Mircea, M., Andreae, M. O., Trebs, I., Hoffer, A., Guyon, P., Artaxo, P., Rizzo, L. V., Lara, L. L., Pauliquevis, T., Maenhaut, W., Raes, N., Chi, X., Mayol-Bracero, O. L., Soto-García, L. L., Claeys, M., Kourtchev, I., Rissler, J., Swietlicki, E., Tagliavini, E., Schkolnik, G., Falkovich, A. H., Rudich, Y., Fisch, G., and Gatti, L. V.: Overview of the inorganic and organic composition of size-segregated aerosol in Rondônia, Brazil, from the biomassburning period to the onset of the wet season, J. Geophys. Res., 112, D01201, https://doi.org/10.1029/2005JD006741, 2007.

Pio, C. A., Legrand, M., Alves, C. A., Oliveira, T., Afonso, J., Caseiro, A., Puxbaum, H., Sanchez-Ochoa, A., and Gelencsér, A.: Chemical composition of atmospheric aerosols during the 2003 summer intense forest fire period, Atmos. Environ., 42, 7530– 7543, https://doi.org/10.1016/j.atmosenv.2008.05.032, 2008.

Samaké, A., Jaffrezo, J.-L., Favez, O., Weber, S., Jacob, V., Albinet, A., Riffault, V., Perdrix, E., Waked, A., Golly, B., Salameh, D., Chevrier, F., Oliveira, D. M., Bonnaire, N., Besombes, J.-L., Martins, J. M. F., Conil, S., Guillaud, G., Mesbah, B., Rocq, B., Robic, P.-Y., Hulin, A., Meur, S. L., Descheemaecker, M., Chretien, E., Marchand, N., and Uzu, G.: Polyols and glucose particulate species as tracers of primary biogenic organic aerosols at 28 French sites, 19, 3357– 3374, https://doi.org/10.5194/acp-19-3357-2019, 2019.

Samaké, A., Bonin, A., Jaffrezo, J.-L., Taberlet, P., Weber, S., Uzu, G., Jacob, V., Conil, S., and Martins, J. M. F.: High levels of primary biogenic organic aerosols are driven by only a few plant-associated microbial taxa, 20, 5609–5628, https://doi.org/10.5194/acp-20-5609-2020, 2020.

**Comment 2#**

**General comments:**

The paper entitled "Saccharide composition in atmospheric fine particulate matter at the remote sites of Southwest China and estimates of source contributions" by Zhenzhen Wang and colleagues provide the characteristic of saccharides during spring 2019 at Lincang, a rural site in Southwest China. The authors reported molecule tracers including anhydrosugars, mono (di) saccharides and sugar alcohols, combined with statistical analysis and HYSPLIT model, they concluded that biofuel and open biomass burning (BB) activities could have a significant impact on ambient aerosol levels at Lincang. Overall, this paper is logically organized, and knowledge of this work is needed and helpful for better understanding air conditions in Southwest China. The topic of this paper is within the scope of the journal Atmospheric Physics and Chemistry. I would like to recommend this paper published after the following of my concerns be resolved.

**Reply**: We appreciate the positive comments and suggestions about the manuscript. We agree with the reviewer's comments, and have updated the manuscript on the basis of these suggestions.

**Major comments:**

1. The surrounding environmental condition is crucial for understanding the results, I strongly suggest the authors added a figure to show the sampling sites as Figure 1. This figure should include some necessary information about the topography, vegetation, residential area nearby Lincang, and photos of three sampling sites are also crucially needed.

   **Reply**: We've added Figure S1 for the location of the sampling sites in the Supporting Information. The number of all the Figures referring to the Supporting Information has been changed.

[Figure]

**Figure S1.** Map of sampling sites. The location of the sampling sites was marked with five-pointed star.

2. The source appointment is mainly based on the 72h backward trajectories of HYSPLIT model. However, high uncertainty existent for the application of HYSPLIT model at high elevation site because topographic relief. The frequencies of HYSPILT or meteorological analysis should provide more creditable results.

   **Reply**: Thank for the reviewer's suggestion. More detailed analyses on topography and meteorology, as well as the frequencies of HYSPILT backward trajectories are stated in the section 3.2 Sources and transport.

   Herein, this sentence has been rewritten in line 472. "46.7% of air mass backward trajectories were generally over 2000 meters, while 53.3% of them were below 2000 meters."

   Some meteorological analysis has been added in line 486-492. "The southwest wind from the Indian Ocean prevailed at Lincang all the year round. In spring, the southwest wind was often affected by the low temperature downhill wind blowing from the snow-

| 221 | | covered Hengduan Mountains. The weather alternated between hot and cold frequently, |
| 222 | | with unstable air pressure and strong wind. Therefore, the lower air could be diluted by |
| 223 | | the relatively clean cold air over the plateau. The upper air mainly came from the |
| 224 | | westerlies." |
| 225 | | |
| 226 | **Minor comments:** | |
| 227 | 1. | The samples of this work are mainly in spring, the title should be changed to "Saccharide |
| 228 | | composition in atmospheric fine particulate matter during spring at the remote sites of |
| 229 | | Southwest China and estimates of source contributions". |
| 230 | | **Reply**: Thank for the reviewer's suggestion. The title have been changed to "Saccharide |
| 231 | | composition in atmospheric fine particulate matter during spring at the remote sites of |
| 232 | | Southwest China and estimates of source contributions". |
| 233 | | |
| 234 | 2. | Line 62, Wu et al., 2020 is not cited in references. |
| 235 | | **Reply**: "Wu et al., 2020" has been corrected to "Wu et al., 2021". "(Wu et al., 2021)" has |
| 236 | | been cited in Line 62 in the revised manuscript. |
| 237 | | |
| 238 | 3. | Line 71-72, "10.1-383.4 ng m$^{-3}$ over the Tibetan Plateau (Li et al., 2019)", the reference |
| 239 | | Li et al., 2019, EP is glacier cryoconites not aerosol samples. |
| 240 | | **Reply**: "10.1-383.4 ng m$^{-3}$ over the Tibetan Plateau (Li et al., 2019)" have been changed |
| 241 | | to "10.1-383.4 ng g$^{-1}$ dry weight in cryoconites over the Tibetan Plateau (Li et al., 2019)". |
| 242 | | |
| 243 | 4. | Line 75, Sichuan Basin, not "Chengdu Basin". |
| 244 | | **Reply**: "Chengdu basin" have been changed to "Sichuan Basin" in line 76. |
| 245 | | |
| 246 | 5. | Line 79-81, Levoglucosan emission of China is estimated by BB activities by Wu et al., |
| 247 | | 2021, this sentence is not rigorous. |
| 248 | | **Reply**: This sentence have been rewritten. "Recently study reported that total |
| 249 | | levoglucosan emission of China exhibited a clear decreasing trend from 2014 (145.7 Gg) |
| 250 | | to 2018 (80.9 Gg) (Wu et al., 2021), suggesting BB activities might reduce in China. |
| 251 | | |
| 252 | 6. | Line 109-112, you should better add some references. |
| 253 | | **Reply**: In line 113, "Referring to the official website of Lincang Municipal People's |
| 254 | | Government, the forest coverage rate of Lincang reaches to 65%." |
| 255 | | |
| 256 | 7. | Line 116, do you have samples over other period? |
| 257 | | **Reply**: We only sampled at the Lincang sites for a period of about a month. |

8.  Line 126-130, please add a figure for sample sites.

    **Reply**: Line 138, we've added Figure S1 for the location of the sampling sites in the Supporting Information.

9.  Line 183, why do not use meteorological data at Lincang?

    **Reply**: The satellite data and Lincang meteorological website data were not exactly the same, but were overall similar. In order to obtain more complete data of all indicators, satellite data were used uniformly.

10. Line 231-233, "no distinct variation", has statistical significance?

    **Reply**: Thank for the reviewer's correction. This sentence is not completely accurate. In the revised manuscript, this sentence was deleted.

11. Line 239-248, samples in those references are not collected at the same period.

    **Reply**: Indeed, the samples in these studies were collected at different times. So we presented the specific sampling time of each research. Even if not all samples were taken in the spring, it would be of great interest to report these information.

12. Line 276-277, how about the L/M for burned ghost money?

    **Reply**: In line 294-298, "It was worth noting that the peak days during 31 March-1 April (L/M = 11.52 ± 1.34) neared the Qingming Festival. Therefore, another possibility of BB events was that people burned large quantities of ghost money, candles and firecrackers to sacrifice ancestor according to Chinese tradition. The main raw materials of ghost money are bamboo and wood."

13. Line 290-291, references for L/$K^+$?

    **Reply**: We've added the references "(Schkolnik et al., 2005; Lee et al., 2010)".

14. Line 431-441, Figure 4, only one air mass from Hengduan Mountain region. Maybe frequency is better for understanding air sources.

    **Reply**: Thank for the reviewer's suggestion. Herein, this sentence has been rewritten in line 472-473. "46.7% of air mass backward trajectories were generally over 2000 meters, while 53.3% of them were below 2000 meters."

15. Line 450-452, how about the atmospheric dynamics for aerosol transport from Southeast Asia to Lincang, especially for residential cooking and heating.

**Reply**: Some sentences were added in line 486-492. "The southwest wind from the Indian Ocean prevailed at Lincang all the year round. In spring, the southwest wind was often affected by the low temperature downhill wind blowing from the snow-covered Hengduan Mountains. The weather alternated between hot and cold frequently, with unstable air pressure and strong wind. Therefore, the lower air could be diluted by the relatively clean cold air over the plateau. The upper air mainly came from the westerlies."

16. Line 512, ng m$^{-3}$?

   **Reply**: In line 561, "µg m$^{-3}$" has been replaced by "ng m$^{-3}$".

17. Line 521, only Myanmar.

   **Reply**: In line 569-571, "The sampling sites suffered from both local emissions and BB via long-range transport from Southeast Asia (Myanmar, Bangladesh) and the northern Indian Peninsula."

**Comment 3**

**General comments:**

   This manuscript presents measurement results of particulate sugar compounds from a rural region in Southwest China. Individual sugar species concentrations, correlations among each other, as well as diagnostic ratios were utilized together with meteorological parameters, back trajectories, and fire counts to constrain the main emission sources, including biomass burning, microorganisms and plant emissions. Biomass burning emissions were the dominant contributor to the ambient PM$_{2.5}$, derived from both local burning activities and long-range transport from surrounding countries.

   The results presented in this paper are interesting as they give insight into the sources of ambient aerosols in this part of China for which limited data have been reported. The results are based on a sound measurement approach, and include a large number of chemical PM components, while the measurement period is relatively short and doesn't show seasonal patterns. Overall, the manuscript is fairly well written and structured, and should therefore be published in ACP following minor revision based on the comments given below.

**Reply**: We appreciate the positive comments and suggestions about the manuscript. We agree with the reviewer's comments, and have updated the manuscript on the basis of these suggestions.

**Specific comments:**

1. It is good to see the utilization of the Metrohm sugar columns (requiring substantially lower eluent concentrations), instead of the usual CarboPak columns from Dionex used in most other studies. Did the authors encounter any co-elution problems of certain sugar species with
this system?

**Reply**: We have encountered some co-elution problems when using the Metrohm sugar
column. At first, we prepared twenty standard saccharide compounds for the method test, and
found that several saccharides co-eluted. By changing the concentration of the eluent and the
flow rate, there were still some saccharides compounds that cannot be separated well.

For example, it was difficult to separate glycerol and sorbitol, the retention times of which
were respectively 5.82 and 5.97 under the condition of the method in this paper. Because
there could be a ~1% deviation of the peak location, data of sorbitol was not accurate and was
not included in this paper. When testing the outfield samples, the sorbitol peak might be
attributed to glycerol.

Under the same condition, we repeated the experiment many times to carefully identify the
peak location for every saccharide. The relative deviation of retention time and peak area
were less than 1%. When it showed a good linear relationship between peak area and
concentration value ($R^2$>99.9%), the saccharides were selected to measure. We finally
decided to test thirteen kinds of saccharide compounds in this article. The selected
saccharides were inositol, glycerol, erythritol, arabitol, trehalose, manitol, mannose, glucose,
fructose, galactosan, levoglucosan, mannosan and sucrose, the retention times of which were
4.88, 5.82, 6.22, 7.84, 8.96, 9.58, 10.93, 11.97, 14.59, 16.94, 17.96, 19.32 and 22.54,
respectively.

Some sentences were added in the section of 2.2 Measurements. "In the preliminary
experiment, some co-elution problems were encountered when using the Metrohm sugar
column. By changing the concentration of the eluent and the flow rate, the measurements of
every saccharide were repeated many times to ensure that the relative deviation of retention
time and peak area was less than 1% and the correlation between peak area and concentration
value was more than 99.9%."

.

2.  Lines 276-278: Do the authors know what are the traditional burning practices during the
Qingming Festival, i.e., what types of biomass the local residents may be burning that are
special for that holiday or is it just enhanced cooking activity, perhaps with more outdoor
BBQ cooking?

**Reply**: The weather around Qingming Day is not very suitable for barbecue. We think the
sudden increase in biomass burining may not be a significant cooking activity. The most
likely activity is the sacrifice around the Tomb-Sweeping Day, during which large quantities
of ghost money, candles and firecrackers were burned. The main raw materials of ghost
money are bamboo and wood.

This sentence has been rewritten in line 294-298. "It was worth noting that the peak days during 31 March-1 April (L/M = 11.52 ± 1.34) neared the Qingming Festival. Therefore, another possibility of BB events was that people burned large quantities of ghost money, candles and firecrackers to sacrifice ancestor according to Chinese tradition. The main raw materials of ghost money are bamboo and wood."

3. Lines 416-418: While erythritol may have been used as surrogate for the 2-methyltetrols, I believe it was mainly for quantification of the 2-methyltetrol peaks when no authentic standards were available, rather than representing the ambient 2-methyltetrol levels. Since the 2-methyltetrols can be separated by HPAEC-PAD, did the authors see any unidentified peaks in the sugar alcohol region of the chromatogram that could potentially be attributed to the 2-methyltetrols?

   **Reply**: The usage of erythritol was due to the lack of the standard 2-methyltetrols. The retention time of erythritol was very short when using the Metrohm sugar columns. The peak positions of erythritol and sorbitol were often overlapped, so it was difficult for us to find other substances in the peak location of the erythritol.

4. Lines 495-500: What are the typical crops that are planted in this region? And what kind of burning practices do the local farmers have, e.g., post-harvest burning of straw or other agricultural residues? Knowledge of these practices would be helpful for explaining the BB patterns and specifically the anhydrosugar diagnostic ratios.

   **Reply**:Thank for the reviewer's suggestion. This region abounds with black tea, nuts, coffee and sugar cane. The main crops in this region are rice, wheat and corn. Crop straw burning is a common phenomenon after the harvest, including the indoor combustion and open burning. We've put these information into the analysis from line 318.

   "Previous results showed the emissions from the combustion of crop residuals such as rice straw, wheat straw and corn straw exhibited comparable $L/K^+$ ratios, typically below 1.0. The averages of $L/K^+$ ratios in this study was $0.48 \pm 0.20$, which was higher than the ratio for wheat straw $(0.10 \pm 0.00)$ and corn straw $(0.21 \pm 0.08)$, but was lower than the ratio for Asian rice straw $(0.62 \pm 0.32)$ (Cheng et al., 2013). In this study, higher $L/K^+$ ratios were observed during 8-10 March $(1.20 \pm 0.19)$ than those during 31 March-1 April $(0.40 \pm 0.13)$, which suggested that the open fire event during 8-10 March was more possibly due to smoldering combustion of residues at low temperatures."

**Technical corrections:**
1. Throughout the manuscript, grammar and wording needs to be polished.

**Reply**:Thank for the reviewer's correction. We'll try the best to polish the grammar and
wording of this manuscript. The writing has been updated with the help of a colleague
scientist whose native language is English.
2.  Lines 144-145: Please, check the correct supplier of the DRI Model 2015 analyzer -- I don't
think that it is "Atmoslytic" anymore but "Magee" or "Aerosol"
**Reply**:We rechecked the relevant information and found that DRI Model 2015 analyzer
was produced by the Aerosol Inc.
Thank for the reviewer's correction. "Atmoslytic Inc." have been changed to "Aerosol
Inc." in line 152.

---

## Author Response (AR4)

**Comment 1**

**General comments:**

In this study the authors reported measurement of $PM_{2.5}$ component over 3 different sites in China during a sampling period of 1 month, during spring 2019. Different saccharides were measured, including biomass burning proxy such as levoglucosan, manossan and galactosan, as well as more uncommon mono(di)saccharide, aiming at tracing the primary biogenic and possibly secondary biogenic sources. After a discussion on the potential link between emissions sources based on correlation and ratio of species, the authors attempt a source-apportionment of the different saccharide using a Non-Negative matrix Factorization (NMF) method and successfully identify 5 different factors of saccharides.

This interesting study reports a comprehensive observational dataset (although not covering the full year) and gives useful insight concerning the sources of organic components thanks to the use of proxy species not-usually used in the literature.

**Reply**:

Dear Prof. Samuel Weber,

We appreciate the positive comments and suggestions about the manuscript. We agree with the reviewer's comments, and have updated the manuscript on the basis of these suggestions.

The grammar and language have been carefully polished. Modifications are listed in **Revision on grammar and language issues** as below. In the marked version of manuscript, the modifications based on the reviewer's comments are in red, the modifications on grammar and language issues are in blue. We also offer a "Certificate of Editing" to certify that this paper has been edited for English language, grammar, punctuation, and spelling by Enago, the editing brand of Crimson Interactive Consulting Co. Ltd.

**Specific comments:**

- Samake et al. (2019) highlight that the different polyols are mostly in the coarse fraction of the PM. Also, it has been hypothesis that the different size distribution of polyols may be a proxy of the different microbiota. Did the authors have also sampled the $PM_{10}$ fraction and could provide the size distribution of the different saccharides?

**Reply:** Thank for the reviewer's suggestion. Indeed, previous results have indicated that polyols (especially mannitol and arabitol) and glucose were prevalent existed in the coarse fraction (Fu et al., 2012; Fuzzi et al., 2007; Pio et al., 2008; Yttri et al., 2007), and were mainly associated with the coarse PM fraction (Samaké et al., 2019). But $PM_{10}$ fraction was not collected due to some practical difficulties, we can't provide the size distribution of the saccharides in this study.

We've cited a reference and rephrased the sentence in line 421-423. "The contribution of fungal spores might be underestimated because previous results had indicated that mannitol and arabitol were mainly associated with the coarse PM fraction (Samaké et al., 2019)."

1. The source apportionment (SA) is a very interesting part, although it lacks of important information that should be reported: Why didn't you included the whole species available in the SA? It could help identify more robustly BB, but also saccharides from soil resuspension (with $Ca^{2+}$), and moreover quantify the apportionment of the different factors to the total $PM_{2.5}$ mass.
   **Reply:** The source apportionment including the other species could quantify the apportionment of the different factors to the total $PM_{2.5}$ mass. We have tried to include the whole species available in the source apportionment. To make the result be better correlate with the five sources of saccharides, we ran a five-factor NMF. The result is shown as below.

[Figure]

Figure 1. The factor profile obtained by NMF analysis based on the saccharide components (a) and the factor profile based on all the species (b).

In Figure 1a, the sources of plant detritus (factor 1), plant senescence (factor 2), biomass burning (factor 3), soil microbiota (factor 4) and airborne pollen (factor 5) respectively contributed 5.3%, 21.0%, 34%, 16.0% and 23.7% to the total saccharides. We matched the factors one-to-one in the two figures according to the characteristic saccharide species. The other various species showed decentralized load on these factors. Based on the compositional data of saccharides, five factors associated to the total $PM_{2.5}$ mass were correspond one-to-one to the factors associated to the total saccharides. Factor 1-4 were correspond to the sources of biomass burning, soil microbiota, plant senescence and airborne pollen, respectively. Factor 5 was more appropriate to be thought as a mixed source.

Thus, in Figure 1b, the sources of biomass burning (factor 1), plant senescence (factor 2), soil microbiota (factor 3), airborne pollen (factor 4) and mix sources

(factor 5) respectively contributed 16.8%, 28.7%, 13%, 15.8% and 25.7% to the total $PM_{2.5}$ mass. However, we think the naming of these factors associated to the total $PM_{2.5}$ mass are not accurate and comprehensive. In order to get more clear information about the sources and their contribution to the total saccharides, we decided to only report the source apportionment of saccharides.

2   It is stated that the SA is still uncertain, but no estimation of the uncertainties is given. It would be of great interest to report the species uncertainties, for instance with bootstraping your input data.
   **Reply:** We only have 91 samples in total, so we cannot carry out resampled runs for many times. The analytical uncertainty was high in present study due to the limited sample number by using the currently used formula in PMF model. We used 0.3 plus the analytical detection limit for estimating uncertainty according to the method of Xie et al. (1999). The constant 0.3 corresponding to the log (geometric standard deviation, GSD) was calculated from the normalized concentrations for all measured species, and was used to represent the variation of measurements. The use of GSD was suitable for our measurement set in a small sample size.

3   The timeserie contribution would also be of great interest. Even if the authors did not include a total variable (namely, $PM_{2.5}$), the timeserie of the total saccharide for the 5 factors would be informative.
   **Reply:** We agree with the reviewer's view of the importance on the timeserie contribution. The timeserie of the total saccharide for the 5 factors are shown in Figure S5. We've rewritten the relevant content from Line 516. "During the sampling periods, daily variations in the proportion of the five factors are shown in Figure S5. Factor 2 soil microbiota emissions could be associated to soil reclamation and cultivation of farming periods, whereas factors 3 plant senescence and factor 5 plant detritus could be associated to the harvesting of vegetation or crops. During the observation period of a month, along with the weather warming as sunshine enhanced, humans left two obvious traces of cultivated soil from 9 to 17 March and from 27 March to 8 April and a trace of vegetation or crop harvest from 17 to 30 March. The stronger pollen discharge occurred in March, probably due to the flowering of certain plants. The BB emissions peaked on 9, 16 March, and 1 April were more prone to be open burnings."

4   The "Soil microbiota" factor, identified mainly by the presence of Trehalose and Mannitol (and Arabitol) denotes with the finding of Samake et al. (2020) that found that Arabitol and Mannitol are associated with fungi and bacteria from the leaves and not with the soil (even if some mixing are probable). I would suggest naming it "Soil and leave microbiota".

**Reply:** We agree with the reviewer's suggestion, "Soil and leaves microbiota" is more specific. We've named it "Soil and leave microbiota" and gave an explanation in line 494-499. "These saccharide compounds had all been detected in the suspended soil particles and associated microbiota (e.g., fungi, bacteria and algae) (Simoneit et al., 2004; Rogge et al., 2007). A recent study found that leaves were a major source of saccharides-associated microbial taxa in a rural area of France (Samaké et al., 2020). Hence, this factor was attributed to soil and leaves microbiota."

5   Overall, the naming of the different factors identified is too rapidly explained, and more detailed could be written to ease the interpretation of the different factors.

**Reply:** Since each type of sugar has been described in the text, the factors were resolved in a little brief way. In the new version, the naming of the different factors has been more detailed explained from Line 489.

"As shown in Figure 6a, factor 1 was characterized by high levels of levoglucosan (71.8%) and mannosan (78.7%), suggesting the source of BB (Simoneit et al., 1999; Nolte et al., 2001). Factor 2 was characterized by trehalose (99.9%) and mannitol (100.0%), and was enriched in the other saccharide components, i.e., arabitol (44.1%), glucose (29.6%), erythritol (18.2%), glycerol (17.8%), levoglucosan (14.7%), and sucrose (8.6%). These saccharide compounds had all been detected in the suspended soil particles and associated microbiota (e.g., fungi, bacteria and algae) (Simoneit et al., 2004; Rogge et al., 2007). A recent study found that leaves were a major source of saccharides-associated microbial taxa in a rural area of France (Samaké et al., 2020). Hence, this factor was attributed to soil and leaves microbiota. Factor 3 had high levels of glycerol (71.4%) and erythritol (58.2%) and showed loadings of glucose (12.8%) and fructose (11.8%). Kang et al. (2018) reported that glycerol and erythritol presented large amounts in winter and autumn when vegetation is decomposed. This factor was attributed to plant senescence and decay by microorganisms. Factor 4 exhibited a predominance of sucrose (78.7%) and showed loadings of glucose (17.2%), arabitol (11.8%). This factor was regarded as the source of airborne pollen, because pollen was the reproductive unit of plants and contains these saccharides and saccharide alcohols as nutritional components (Bieleski, 1995; Miguel et al., 2006; Fu et al., 2012). Factor 5 characterized by the dominance of fructose (88.2%) was resolved, and was enriched in glucose (38.2%) and arabitol (21.2%), thus it could be regarded as the source of plant detritus."

**Minor comment:**
1   Please provide the pie chart of Figure 6b in a non-3D way, as the relative proportion is much harder to see in 3D compare to regular 2D graph.

**Reply:** We agree with the reviewer's comment. We've provided the pie chart of Figure 6b in a 2D way in the new version of manuscript.

[Figure]

**Figure 6**. Factor profile obtained by NMF analysis (a). Source contribution of the five factors to the total saccharides in PM$_{2.5}$ samples (b).

**References:**

Fuzzi, S., Decesari, S., Facchini, M. C., Cavalli, F., Emblico, L., Mircea, M., Andreae, M. O., Trebs, I., Hoffer, A., Guyon, P., Artaxo, P., Rizzo, L. V., Lara, L. L., Pauliquevis, T., Maenhaut, W., Raes, N., Chi, X., Mayol-Bracero, O. L., Soto-García, L. L., Claeys, M., Kourtchev, I., Rissler, J., Swietlicki, E., Tagliavini, E., Schkolnik, G., Falkovich, A. H., Rudich, Y., Fisch, G., and Gatti, L. V.: Overview of the inorganic and organic composition of size-segregated aerosol in Rondônia, Brazil, from the biomassburning period to the onset of the wet season, J. Geophys. Res., 112, D01201, https://doi.org/10.1029/2005JD006741, 2007.

Pio, C. A., Legrand, M., Alves, C. A., Oliveira, T., Afonso, J., Caseiro, A., Puxbaum, H., Sanchez-Ochoa, A., and Gelencsér, A.: Chemical composition of atmospheric aerosols during the 2003 summer intense forest fire period, Atmos. Environ., 42, 7530–7543, https://doi.org/10.1016/j.atmosenv.2008.05.032, 2008.

Samaké, A., Jaffrezo, J.-L., Favez, O., Weber, S., Jacob, V., Albinet, A., Riffault, V., Perdrix, E., Waked, A., Golly, B., Salameh, D., Chevrier, F., Oliveira, D. M., Bonnaire, N., Besombes, J.-L., Martins, J. M. F., Conil, S., Guillaud, G., Mesbah, B., Rocq, B., Robic, P.-Y., Hulin, A., Meur, S. L., Descheemaecker, M., Chretien, E., Marchand, N., and Uzu, G.: Polyols and glucose particulate species as tracers of primary biogenic organic aerosols at 28 French sites, 19, 3357–3374, https://doi.org/10.5194/acp-19-3357-2019, 2019.

Samaké, A., Bonin, A., Jaffrezo, J.-L., Taberlet, P., Weber, S., Uzu, G., Jacob, V., Conil, S., and Martins, J. M. F.: High levels of primary biogenic organic aerosols are driven by only a few plant-associated microbial taxa, 20, 5609–5628, https://doi.org/10.5194/acp-20-5609-2020, 2020.

**Comment 2**

**General comments:**

The paper entitled "Saccharide composition in atmospheric fine particulate matter at the remote sites of Southwest China and estimates of source contributions" by Zhenzhen Wang and colleagues provide the characteristic of saccharides during spring 2019 at Lincang, a rural site in Southwest China. The authors reported molecule tracers including anhydrosugars, mono (di) saccharides and sugar alcohols, combined with statistical analysis and HYSPLIT model, they concluded that biofuel and open biomass burning (BB) activities could have a significant impact on ambient aerosol levels at Lincang. Overall, this paper is logically organized, and knowledge of this work is needed and helpful for better understanding air conditions in Southwest China. The topic of this paper is within the scope of the journal Atmospheric Physics and Chemistry. I would like to recommend this paper published after the following of my concerns be resolved.

**Reply**: We appreciate the positive comments and suggestions about the manuscript. We agree with the reviewer's comments, and have updated the manuscript on the basis of these suggestions.

**Major comments:**

1. The surrounding environmental condition is crucial for understanding the results, I strongly suggest the authors added a figure to show the sampling sites as Figure 1. This figure should include some necessary information about the topography, vegetation, residential area nearby Lincang, and photos of three sampling sites are also crucially needed.

    **Reply**: We've added Figure S1 for the location of the sampling sites in the Supporting Information. The number of all the Figures referring to the Supporting Information has been changed.

[Figure]

**Figure S1.** Map of sampling sites. The location of the sampling sites is marked with

five-pointed star.

2. The source appointment is mainly based on the 72h backward trajectories of HYSPLIT model. However, high uncertainty existent for the application of HYSPLIT model at high elevation site because topographic relief. The frequencies of HYSPILT or meteorological analysis should provide more creditable results.

    **Reply**: Thank for the reviewer's suggestion. More detailed analyses on topography and meteorology, as well as the frequencies of HYSPILT backward trajectories are stated in the section 3.2 Sources and transport.

    Herein, this sentence has been rewritten. "51.6% of air mass backward trajectories were generally above 2000 meters, whereas 48.4% of them were below 2000 meters."

    "The southwest wind from the Indian Ocean prevailed at Lincang all year-round. In spring, the southwest wind was often affected by the low temperature downhill wind blowing from the snow-covered Hengduan Mountains. The weather frequently alternated between hot and cold, with unstable air pressure and strong wind. Therefore, the lower air could be diluted by the relatively clean cold air over the plateau. The upper air mainly came from the westerlies."

**Minor comments:**

1. The samples of this work are mainly in spring, the title should be changed to "Saccharide composition in atmospheric fine particulate matter during spring at the remote sites of Southwest China and estimates of source contributions".

    **Reply**: Thank for the reviewer's suggestion. The title have been changed to "Saccharide composition in atmospheric fine particulate matter during spring at the remote sites of Southwest China and estimates of source contributions".

2. Line 62, Wu et al., 2020 is not cited in references.

    **Reply**: Wu et al., 2020 have been cited in Line 62 in the revised manuscript.

3. Line 71-72, "10.1-383.4 ng m$^{-3}$ over the Tibetan Plateau (Li et al., 2019)", the reference Li et al., 2019, EP is glacier cryoconites not aerosol samples.

**Reply**: "10.1-383.4 ng m$^{-3}$ over the Tibetan Plateau (Li et al., 2019)" have been changed to "10.1-383.4 ng g$^{-1}$ dry weight in cryoconites over the Tibetan Plateau (Li et al., 2019)".

4. Line 75, Sichuan Basin, not "Chengdu Basin".
   **Reply**: "Chengdu basin" have been changed to "Chengdu plain".

5. Line 79-81, Levoglucosan emission of China is estimated by BB activities by Wu et al., 2021, this sentence is not rigorous.
   **Reply**: This sentence have been rewritten. "Recently study reported that total levoglucosan emission of China exhibited a clear decreasing trend from 2014 (145.7 Gg) to 2018 (80.9 Gg) (Wu et al., 2021), suggesting BB activities might reduce in China.

6. Line 109-112, you should better add some references.
   **Reply**: "Referring to the official website of Lincang Municipal People's Government, the forest coverage rate of Lincang reaches to 65%."

7. Line 116, do you have samples over other period?
   **Reply**: We only sampled at the Lincang sites for a period of about a month.

8. Line 126-130, please add a figure for sample sites.
   **Reply**: We've added Figure S1 for the location of the sampling sites in the Supporting Information.

9. Line 183, why do not use meteorological data at Lincang?
   **Reply**: The satellite data and Lincang meteorological website data were not exactly the same, but were overall similar. In order to obtain more complete data of all indicators, satellite data were used uniformly.

10. Line 231-233, "no distinct variation", has statistical significance?
    **Reply**: Thank for the reviewer's correction. This sentence is not completely accurate. In the revised manuscript, this sentence was deleted.

11. Line 239-248, samples in those references are not collected at the same period.

**Reply**: Indeed, the samples in these studies were collected at different times. So we presented the specific sampling time of each research. Even if not all samples were taken in the spring, it would be of great interest to report these information.

12. Line 276-277, how about the L/M for burned ghost money?

    **Reply**: "It was worth noting that the peak days during 31 March-1 April (L/M = 11.52 ± 1.34) neared the Qingming Festival. Another possibility of BB events was that people burned ghost money to sacrifice ancestor according to Chinese tradition."

13. Line 290-291, references for $L/K^+$?

    **Reply**: We've added the references "(Schkolnik et al., 2005; Lee et al., 2010)".

14. Line 431-441, Figure 4, only one air mass from Hengduan Mountain region. Maybe frequency is better for understanding air sources.

    **Reply**: Thank for the reviewer's suggestion. Herein, this sentence has been rewritten. "51.6% of air mass backward trajectories were generally above 2000 meters, whereas 48.4% of them were below 2000 meters."

15. Line 450-452, how about the atmospheric dynamics for aerosol transport from Southeast Asia to Lincang, especially for residential cooking and heating.

    **Reply**: Some sentences were added. "The southwest wind from the Indian Ocean prevailed at Lincang all year-round. In spring, the southwest wind was often affected by the low temperature downhill wind blowing from the snow-covered Hengduan Mountains. The weather frequently alternated between hot and cold, with unstable air pressure and strong wind. Therefore, the lower air could be diluted by the relatively clean cold air over the plateau. The upper air mainly came from the westerlies."

16. Line 512, ng m$^{-3}$?

    **Reply**: "μg m$^{-3}$" has been replaced by "ng m$^{-3}$".

17. Line 521, only Myanmar.

**Reply**: "The sampling sites suffered from both local emissions and BB via long-range transport from Southeast Asia (Myanmar, Bangladesh) and the northern Indian Peninsula."

**Comment 3**

**General comments:**

This manuscript presents measurement results of particulate sugar compounds from a rural region in Southwest China. Individual sugar species concentrations, correlations among each other, as well as diagnostic ratios were utilized together with meteorological parameters, back trajectories, and fire counts to constrain the main emission sources, including biomass burning, microorganisms and plant emissions. Biomass burning emissions were the dominant contributor to the ambient $PM_{2.5}$, derived from both local burning activities and long-range transport from surrounding countries.

The results presented in this paper are interesting as they give insight into the sources of ambient aerosols in this part of China for which limited data have been reported. The results are based on a sound measurement approach, and include a large number of chemical PM components, while the measurement period is relatively short and doesn't show seasonal patterns. Overall, the manuscript is fairly well written and structured, and should therefore be published in ACP following minor revision based on the comments given below.

**Reply**: We appreciate the positive comments and suggestions about the manuscript. We agree with the reviewer's comments, and have updated the manuscript on the basis of these suggestions.

**Specific comments:**

1. It is good to see the utilization of the Metrohm sugar columns (requiring substantially lower eluent concentrations), instead of the usual CarboPak columns from Dionex used in most other studies. Did the authors encounter any co-elution problems of certain sugar species with this system?

   **Reply**: We have encountered some co-elution problems when using the Metrohm sugar column. At first, we prepared twenty standard saccharide compounds for the method test, and found that several saccharides co-eluted. By changing the concentration of the eluent and the flow rate, there were still some saccharides compounds that cannot be separated well.

For example, it was difficult to separate glycerol and sorbitol, the retention times of which were respectively 5.82 and 5.97 under the condition of the method in this paper. Because there could be a ~5% deviation of the peak location, data of sorbitol was not accurate and was not included in this paper. When testing the outfield samples, the sorbitol peak might be attributed to glycerol.

Under the same condition, we repeated the experiment many times to carefully identify the peak location for every saccharide. The relative deviation of retention time and peak area were less than 1%. When it showed a good linear relationship between peak area and concentration value ($R^2$>99.9%), the saccharides were selected to measure. We finally decided to test thirteen kinds of saccharide compounds in this article. The selected saccharides were inositol, glycerol, erythritol, arabitol, trehalose, manitol, mannose, glucose, fructose, galactosan, levoglucosan, mannosan and sucrose, the retention times of which were 4.88, 5.82, 6.22, 7.84, 8.96, 9.58, 10.93, 11.97, 14.59, 16.94, 17.96, 19.32 and 22.54, respectively.

2. Lines 276-278: Do the authors know what are the traditional burning practices during the Qingming Festival, i.e., what types of biomass the local residents may be burning that are special for that holiday or is it just enhanced cooking activity, perhaps with more outdoor BBQ cooking?
   **Reply**: The weather around Qingming Day is not very suitable for barbecue. We think the sudden increase in biomass burining may not be a significant cooking activity. The most likely activity is the sacrifice around the Tomb-Sweeping Day, during which large quantities of ghost money, candles and firecrackers were burned. The main raw materials of ghost money are bamboo and wood.

3. Lines 416-418: While erythritol may have been used as surrogate for the 2-methyltetrols, I believe it was mainly for quantification of the 2-methyltetrol peaks when no authentic standards were available, rather than representing the ambient 2-methyltetrol levels. Since the 2-methyltetrols can be separated by HPAEC-PAD, did the authors see any unidentified peaks in the sugar alcohol region of the chromatogram that could potentially be attributed to the 2-methyltetrols?
   **Reply**: The usage of erythritol was due to the lack of the standard 2-methyltetrols. The retention time of erythritol was very short when using the Metrohm sugar

columns. The peak positions of erythritol and sorbitol were often overlapped, so it was difficult for us to find other substances in the peak location of the erythritol.

4. Lines 495-500: What are the typical crops that are planted in this region?   And what kind of burning practices do the local farmers have, e.g., post-harvest burning of straw or other agricultural residues? Knowledge of these practices would be helpful for explaining the BB patterns and specifically the anhydrosugar diagnostic ratios.

   **Reply**:Thank for the reviewer's suggestion. This region abounds with black tea, nuts, coffee and sugar cane. The main crops in this region are rice, wheat and corn. Crop straw burning is a common phenomenon after the harvest, including the indoor combustion and open burning. We've put these information into the analysis. "Previous results showed the emissions from the combustion of crop residuals such as rice straw, wheat straw and corn straw exhibited comparable $L/K^+$ ratios, typically below 1.0. The averages of $L/K^+$ ratios in this study was $0.48 \pm 0.20$, which was higher than the ratio for wheat straw ($0.10 \pm 0.00$) and corn straw ($0.21 \pm 0.08$), but was lower than the ratio for Asian rice straw ($0.62 \pm 0.32$) (Cheng et al., 2013). In this study, higher $L/K^+$ ratios were observed during 8-10 March ($1.20 \pm 0.19$) than those during 31 March-1 April ($0.40 \pm 0.13$), which suggested that the open fire event during 8-10 March was more possibly due to smoldering combustion of residues at low temperatures."

**Technical corrections:**

1. Throughout the manuscript, grammar and wording needs to be polished.

**Reply**:Thank for the reviewer's correction. We'll try the best to polish the grammar and wording of this manuscript. The writing has been updated with the help of a colleague scientist whose native language is English. The specific changes on grammar and wording are listed in **Revision on grammar and language issues** as below.

2. Lines 144-145: Please, check the correct supplier of the DRI Model 2015 analyzer -- I don't think that it is "Atmoslytic" anymore but "Magee" or "Aerosol"

   **Reply**:   We rechecked the relevant information and found that DRI Model 2015 analyzer was produced by the Aerosol Inc.

   Thank for the reviewer's correction. "Atmoslytic Inc." have been changed to "Aerosol Inc."

**Revision on grammar and language issues**

In the marked manuscript, in addition to the above modifications in red, the following modifications on grammar and language issues have been made in blue.

1. Line 18, "characteristic" has been changed to "characteristics".
2. Line 19, "was" has been replaced by "were".
3. Line 21, "OC" has been changed to "organic compound (OC)".
4. Line 29, "as" has been deleted.
5. Line 32, "the contribution" has been replaced by "contributions".
6. Line 33, "the" has been deleted.
7. Line 41, the sentence has been rewritten. "Atmospheric saccharide components have been extensively reported to originate from natural or anthropogenic biomass burning (BB), suspended soil or dust and primary biological aerosol particles (PBAPs), e.g., fungal and fern spores, pollens, algae, fungi, bacteria, and plant debris, and biogenic secondary organic aerosol (SOA) (e.g., Rogge et al., 1993; Graham et al., 2003; Jaenicke, 2005; Medeiros et al., 2006; Elbert et al., 2007; Fu et al., 2013)."
8. Line 47, "classes" has been replaced by "class".
9. Line 52, ", and as much as" has been replaced by "and up to".
10. Line 60, "not suitable" has been replaced by "unsuitable".
11. Line 61, "is" has been replaced by "was".
12. Line 72, "the" has been added.
13. Line 73, "concentrations" has been replaced by "concentration". "is" has been replaced by "was". "the" has been added.
14. Line 77, the sentence has been rewritten. "In general, BB with a notable contribution to OC was an important source of fine particulate matter in China (Zhang et al., 2008; Cheng et al., 2013; Chen et al., 2017)."
15. Line 87, "and plant as well as animal debris" has been replaced by "as well as plant and animal debris".
16. Line 100, "can comprise from 20-30%" has been replaced by "could account for 20%–30%".
17. Line 102, the sentence has been rewritten. "However, studies on quantifying the abovementioned biogenic aerosol contributions to ambient aerosol are inadequate."
18. Line 105, "Lincang located on the southwest border of China" has been replaced by "Lincang, located on the southwest border of China,".
19. Line 112, the sentence has been rewritten. "The proportion of houses that employ

wood burning for cooking is very high in villages in proximity and a large area of Southeast Asia, and forest fires frequently occur in this area, especially in the dry seasons (March–April)."

20. Line 115, the sentence has been rewritten. "These imply that there are abundant biogenic aerosols in Lincang, and BB pollution may be an essential potential source of air pollution."

21. Line 119, "and BB types" has been replaced by "as well as BB types,".

22. Line 120, "were" has been replaced by "was". "March 8 to April 8" has been replaced by "8 March to 9 April".

23. Line 122, "(including anhydrosugars and $K^+$), and" has been replaced by ", including anhydrosugars and $K^+$, as well as".

24. Line 125, "provide" has been replaced by "providing".

25. Line 130, the sentence has been rewritten. "$PM_{2.5}$ samples were simultaneously collected on three mountaintop sites in Lincang, respectively of Datian (24.11◦ N, 100.13◦ E, 1960 m asl), Dashu (24.12◦ N, 100.11◦ E, 1840 m asl) and Yakoutian (24.12◦ N, 100.09◦ E, 1220 m asl), which are located ~300 km west of Kunming (the capital of Yunnan province in China) and ~120 km east from the Burma border (shown in Figure S1)."

26. Line 139, "during 8 March to 9 April in 2019" has been deleted.

27. Line 148, "organic carbon (OC)" has been replaced by "OC".

28. Line 156, "10.0 mL of de-ionized" has been replaced by "10.0 mL deionized".

29. Line 165, "high performance anion-exchange chromatography system coupled with a pulsed amperometric detector (HPAEC-PAD)" has been replaced by "high-performance anion-exchange chromatography coupled with a pulsed amperometric detector".

30. Line 173, "a Metrosep Carb 2-250 analytical column" has been replaced by "a Metrosep Carb 4-250 analytical column".

31. Line 199, "Statistical Product and Service Solutions (SPSS)" has been replaced by "statistical product and service solutions".

32. Line 212, "less" has been replaced by "fewer".

33. Line 214, the sentence has been rewritten. "In this study, galactosan, mannose and inositol were excluded because their concentration in most samples was below the DL."

34. Line 215, the sentence has been rewritten. "Concentrations of the other ten saccharide species for a total of 91 samples were subjected to NMF analysis."

35. Line 218, "Geometric Standard Deviation, GSD" has been replaced by "geometric standard deviation".

36. Line 221, "Discussion" has been replaced by "discussion".

37. Line 224, "are" has been replaced by "is".

38. Line 225, the sentence has been rewritten. "During the sampling periods, the $PM_{2.5}$ mass concentrations ranged between 13.7 and 87.8 µg m$^{-3}$ with an average value of 41.8 µg m$^{-3}$. The concentrations of OC and EC, respectively, were in the range of 2.5–22.4 and 0.3–4.3 µg m$^{-3}$ with average values of 8.4 and 1.7 µg m$^{-3}$."

39. Line 229, the sentence has been rewritten. "The ambient concentrations of the total saccharides varied between 244.5 and 1291.6 ng m$^{-3}$ with an average value of 638.4 ng m$^{-3}$."

40. Line 233, "mean" has been replaced by "average". The "mean" elsewhere in the article have all been replaced by "average".

41. Line 234, "sugar" has been replaced by "mono (di) saccharide".

42. Line 235, "the" has been deleted.

43. Line 240, the sentence has been rewritten. "The average concentrations of levoglucosan and mannosan were 287.7 and 31.6 ng m$^{-3}$, respectively, with respective ranges of 95.6–714.7 and 0–134.7 ng m$^{-3}$ for all 91 samples. Galactosan was detected only in six samples, with a range of 2.5–5.5 ng m$^{-3}$. The anhydrosugars accounted for 48.5% of the total measured saccharides."

44. Line 255, "March 8, 16, 23 and April 1" has been replaced by "8, 16, 23 March and 1 April". "that" has been added.

45. Line 256, "were" has been replaced by "was".

46. Line 265, "the" has been added.

47. Line 270, "ranging" has been replaced by "ranged".

48. Line 275, "during 31 March-1 April and during 8-10 March" has been replaced by "from 31 March to 1 April and from 8 to 10 March". All relevant questions have been revised elsewhere in the manuscript.

49. Line 281, "whilst" has been replaced by "while".

50. Line 283, the sentence has been rewritten. "A possibility of BB events was that people burned ghost money as sacrifice to their ancestors according to Chinese tradition."

51. Line 302, "averages" has been replaced by "average".

52. Line 313, the sentence has been rewritten. "It showed that $K^+$ was more highly correlated with $PM_{2.5}$, OC, and EC, which could be explained by either the photooxidative decay of levoglucosan (Hennigan et al., 2010) and/or different types of BB processes (Schkolnik et al., 2005; Lee et al., 2010)."

53. Line 318, "is" has been replaced by "was".

54. Line 320, "the" has been added.

55. Line 323, the sentence has been rewritten. "However, certainly, it was not from coal burning (0.0001–0.001) and waste incineration (0.0022)."

56. Line 325, "19.1 to 81.3%" has been replaced by "19.1%–73.9%". "are" has been replaced by "were".

57. Line 335, "However" has been deleted.

58. Line 339, "(Wu et al., 2020)" has been replaced by "(Wu et al., 2021)".

59. Line 340, "and biomass burning activities have been reduced" has been deleted.

60. Line 341, the sentence has been rewritten. "However, it was noteworthy that the average concentration of levoglucosan (287.7 ng m$^{-3}$) and the BB contributions to OC (41.3%) at Lincang mountain site were both higher than the values of 191.8 ng m$^{-3}$ and 28.4% at Tengchong mountain site in 2004 spring (Sang et al., 2013). The result suggested that no significant reduction in BB emissions in Southwest Yunnan Province."

61. Line 349, "were 25.2-373.7 ng m$^{-3}$" has been replaced by "were in the range of 25.2–373.7 ng m$^{-3}$".

62. Line 353, the sentence has been rewritten. "The results agreed with those of previous studies (Yttri et al., 2007; Jia et al., 2010; Fu et al., 2012), which had found that sucrose was one of the dominate specie in spring fine aerosols."

63. Line 356, "in particular of spring blossom season" has been replaced by ", especially in the spring blossom season".

64. Line 358, "the" has been deleted.

65. Line 370, "were" has been replaced by "was".

66. Line 390, the sentence has been rewritten. "These reduced sugars are often reported to be related to plant senescence and decay by microorganisms (Simoneit et al., 2004; Tsai et al., 2013); and they are produced by fungi, lichens, soil biota and algae (Elbert et al., 2007; Bauer et al., 2008)."

67. Line 392, the sentence has been rewritten. "The average concentration of total sugar alcohols was 159.9 ng m$^{-3}$ with a range of 53.1−254.0 ng m$^{-3}$, which accounted for 26.6 ± 9.9% of the total measured saccharides."

68. Line 395, "Previous studies suggested that the source of glycerol was not be specific to biological emissions, biomass combustion might increase atmospheric glycerol

concentrations (Jia et al., 2010; Graham et al., 2002; Wang et al., 2011)." has been added.

69. Line 398, the sentence has been rewritten. "Herein, glycerol was the second most abundant saccharide, with an average concentration of 123.7 ng m$^{-3}$ accounting for 5.1%−44.6% (average: 22.6%) of the total measured saccharides."

70. Line 410, "correlations" has been replaced by "correlation".

71. Line 416, "can" has been replaced by "could".

72. Line 432, the sentence has been rewritten. "In this study, the concentration range of erythritol was 0.4–19.8 ng m$^{-3}$ (average: 11.1 ng m$^{-3}$)."

73. Line 435, "originates" has been replaced by "originate". "derives" has been deleted.

74. Line 437, "the" has been added.

75. Line 440, the sentence has been rewritten. "In this study, only inositol correlated with levoglucosan (R = 0.42), suggesting that inositol might be linked to biomass combustion sources."

76. Line 442, "primarily" has been replaced by "primary".

77. Line 445, the sentence has been rewritten. "Since the distinct concentration of the studied compounds was due to different emission sources arising from different wind directions, the 72 h backward trajectories for the samples at the Dashu site (24.12◦ N, 100.11◦ E) and the spatial distribution of the fire spots (8 March to 9 April, 2019) were calculated to understand the source of saccharides in aerosol (Figure 4)."

78. Line 452, "over 2000 meters" has been replaced by "above 2000 meters". All "over 2000 meters" has been replaced by "above 2000 meters" in the whole article. Therefore, the "Figure 5" has been changed to a new version.

[Figure]

79. Line 460, the sentence has been rewritten. "The average concentrations of saccharide compounds, as well as the contribution of them, for the episodes above and below 2000 meters are shown in Figure 5. The average concentrations of

levoglucosan and mannosan for the above 2000 meters samples (327.4 and 35.6 ng m$^{-3}$) were higher than those for the below 2000 meters samples (250.3 and 27.3 ng m$^{-3}$). The anhydrosugars accounted for 49.2% and 36.9% of total saccharides, respectively for the above and below 2000 meters samples."

80. Line 472, the sentence has been rewritten. "These results were in agreement with the fact that residents across Southeast Asia use wood as an energy source to cook and generate heat."

81. Line 475, the sentence has been rewritten. "While for glucose, fructose and sucrose, it was a little higher in the below 2000 meters samples (mean 33.5, 26.4, and 106.2 ng m$^{-3}$) than that in the above 2000 meters samples (mean 29.2, 22.9, and 67.8 ng m$^{-3}$)."

82. Line 481, "pollen" has been replaced by "pollens".

83. Line 488, "to" has been replaced by "with".

84. Line 492, "saccharides" has been replaced by "saccharide".

85. Line 499, "has" has been replaced by "had".

86. Line 502, the sentence has been rewritten. "This factor was attributed to plant senescence and decay by microorganisms."

87. Line 505, "is" has been replaced by "was".

88. Line 511, "charts" has been replaced by "chart".

89. Line 513, "Factor" has been replaced by "Factors".

90. Line 516, "daily variations on proportion" has been replaced by "daily variations in the proportion".

91. Line 519, "and" has been replaced by "whereas".

92. Line 520, "the" has been added. "crop" has been replaced by "crops".

93. Line 521, "human" has been replaced by "humans".

94. Line 533, "the" has been deleted.

95. Line 534, "characteristic" has been replaced by "characteristics".

96. Line 536, "the" has been replaced by "a".

97. Line 540, "appeared to be" has been replaced by "are".

98. Line 550 "herein" and "the" have been deleted.

99. Line 552, "field burnings of agricultural residues and fallen leaves, as well as forest fire" has been replaced by "field burning of agricultural residues, fallen leaves, and forest fire".

100. In the section of "**Captions of Figure and Table**", some corrections have been made accordingly.

[Figure]

Author First, Quality First
www.enago.cn

**CERTIFICATE OF EDITING**

This is to certify that the paper titled Saccharide composition in atmospheric fine particulate matter during spring at the remote sites of Southwest China and estimates of source contributions commissioned to us by Zhenzhen has been edited for English language, grammar, punctuation, and spelling by Enago, the editing brand of Crimson Interactive Consulting Co. Ltd. under Normal Editing B2C.

✓ **ISO 17100:2015**
Translation Service
Providers

✓ **ISO 27001:2013**
Information Security
Management System

✓ **ISO 9001:2015**
Quality Management
System

Issued by:
Enago, Crimson Interactive (Beijing) Consulting Co., Ltd.
Room 607, Zhucheng Building,
No. 6 Zhongguancun South Street,
Haidian District, Beijing

Disclaimer : The intent of the author's message has been preserved during the editing process. The author is free to accept or reject our changes in the document after reviewing our edits. This certificate has been awarded at the time of sharing the final edited version (full file or sections of the file) with the author. Enago does not bear any responsibility for any alterations done by the author to the edited document post 1 Jul 2021.

| | | | |
|---|---|---|---|
| **Japan** | www.enago.jp, www.ulatus.jp, www.voxtab.jp | **Russia** | www.enago.ru |
| **Taiwan** | www.enago.tw, www.ulatus.tw | **Arabic** | www.enago.ae |
| **China** | www.enago.cn, www.ulatus.cn | **Turkey** | www.enago.com.tr |
| **Brazil** | www.enago.com.br, www.ulatus.com.br | **S. Korea** | www.enago.co.kr |
| **Germany** | www.enago.de | **Global** | www.enago.com, www.ulatus.com, www.voxtab.com |

**About Crimson:**
Crimson Interactive Consulting Co. Ltd. is one of the world's leading academic research support services. Since 2005, we've supported over 2 million researchers in 125 countries with their publication goals.